# Anthraquinones, Diphenyl Ethers, and Their Derivatives from the Culture of the Marine Sponge-Associated Fungus *Neosartorya spinosa* KUFA 1047 [note 1]

**DOI:** 10.3390/md19080457

**Published:** 2021-08-11

**Authors:** Joana D. M. de Sá, José A. Pereira, Tida Dethoup, Honorina Cidade, Maria Emília Sousa, Inês C. Rodrigues, Paulo M. Costa, Sharad Mistry, Artur M. S. Silva, Anake Kijjoa

**Affiliations:** 1Laboratório de Química Orgânica, Departamento de Ciências Químicas, Faculdade de Farmácia, Universidade do Porto, Rua de Jorge Viterbo Ferreira, 228, 4050-313 Porto, Portugal; joanadmsa2703@gmail.com (J.D.M.d.S.); hcidade@ff.up.pt (H.C.); esousa@ff.up.pt (M.E.S.); 2ICBAS—Instituto de Ciências Biomédicas Abel Salazar, Rua de Jorge Viterbo Ferreira, 228, 4050-313 Porto, Portugal; jpereira@icbas.up.pt (J.A.P.); inescoutorodrigues@gmail.com (I.C.R.); pmcosta@icbas.up.pt (P.M.C.); 3Interdisciplinary Centre of Marine and Environmental Research (CIIMAR), Terminal de Cruzeiros do Porto de Lexões, Av. General Norton de Matos s/n, 4450-208 Matosinhos, Portugal; 4Department of Plant Pathology, Faculty of Agriculture, Kasetsart University, Bangkok 10240, Thailand; tdethoup@yahoo.com; 5Department of Chemistry, University of Leicester, University Road, Leicester LE 7RH, UK; scm11@leicester.ac.uk; 6Departamento de Química & QOPNA, Universidade de Aveiro, 3810-193 Aveiro, Portugal; artur.silva@ua.pt

**Keywords:** *Neosartorya spinosa*, Trichocomaceae, marine sponge-associated fungus, anthraquinones, biphenyl ethers, anti-tyrosinase, antibacterial activity, antibiofilm activity

## Abstract

Previously unreported anthraquinone, acetylpenipurdin A (**4**), biphenyl ether, neospinosic acid (**6**), dibenzodioxepinone, and spinolactone (**7**) were isolated, together with (*R*)-6-hydroxymellein (**1**), penipurdin A (**2**), acetylquestinol (**3**), tenellic acid C (**5**), and vermixocin A (**8**) from the culture of a marine sponge-associated fungus *Neosartorya spinosa* KUFA1047. The structures of the previously unreported compounds were established based on an extensive analysis of 1D and 2D NMR spectra as well as HRMS data. The absolute configurations of the stereogenic centers of **5** and **7** were established unambiguously by comparing their calculated and experimental electronic circular dichroism (ECD) spectra. Compounds **2** and **5**–**8** were tested for their in vitro acetylcholinesterase and tyrosinase inhibitory activities as well as their antibacterial activity against Gram-positive and Gram-negative reference, and multidrug-resistant strains isolated from the environment. The tested compounds were also evaluated for their capacity to inhibit biofilm formation in the reference strains.

## 1. Introduction

Fungi are among organisms that have a remarkable capacity to produce different classes of structurally diverse secondary metabolites with relevant biological and pharmacological activities. This capability may be due to their necessity to produce highly bioactive molecules for their communications or inhibition of the growth of antagonistic neighbor microorganisms with which they cohabit in the same ecological niches [1]. Although secondary metabolites of terrestrial fungi have been extensively investigated for many decades due to their importance in pharmaceutical research [2], only in the past two decades that marine-derived fungi started to gain more attention from researchers [3]. Marine-derived fungi have become one of the most important sources of bioactive compounds not only because they are among the world’s most important resources for unprecedented chemodiversity but also because they can produce quantity of compounds with potential for drug development, clinical trials, and even marketing [4].

In our research program with an objective to search for new bioactive compounds from marine-derived fungi, we investigated several members of the genus *Neosartorya* (Trichocomaceae) isolated from different marine organisms such as sponges, coral, and algae. Many different chemical classes of secondary metabolites, such as polyketides, isocoumarins, ergosterol derivatives, meroditerpenes, pyripyropenes, benzoic acid derivatives, prenylated indole derivatives, tryptoquivalines, fiscalins, phenylalanine-derived alkaloids, and cyclopeptides, have been isolated and investigated for their anticancer and antibacterial activities [5,6,7]. Therefore, in our ongoing search for new natural antibiotics from marine-derived fungi, we investigated secondary metabolites from the culture of *N. spinosa* KUFA 1047, isolated from a marine sponge *Mycale* sp., which was collected from the Samae San Island in the Gulf of Thailand. Although the soil-derived *N. spinosa* has already been investigated for its secondary metabolites [8], this is the first study of the secondary metabolites from a marine-derived *N. spinosa*.

Fractionation of the ethyl acetate extract of the culture of *N. spinosa* KUFA 1047 by column chromatography of silica gel, followed by purification by preparative TLC, Sephadex LH-20 column, and crystallization led to the isolation of undescribed acetylpenipurdin A (**4**), neospinosic acid (**6**), and spinolactone (**7**), as well as the previously reported (*R*)-6-hydroxymellein (**1**) [9,10], penipurdin A (**2**) [11], acetylquestinol (**3**) [12], tenellic acid C (**5**) [13], and vermixocin A (**8**) [14,15,16] (Figure 1). The structures of the undescribed compounds were established based on an extensive analysis of their 1D and 2D NMR as well as HRMS spectra. In the case of **5** and **7**, the absolute configurations of their stereogenic carbons were established by comparison of their experimental and calculated electronic circular dichroism (ECD) spectra.

Compounds **2** and **5**–**8** were tested for their in vitro acetylcholinesterase and tyrosinase inhibitory activities, as well as their antibacterial activity against Gram-negative and Gram-positive bacteria by disk diffusion and by determination of the minimum inhibitory concentration (MIC) and minimal bactericidal concentration (MBC) of several reference strains and environmental multidrug-resistant isolates. The tested compounds were also evaluated for their potential synergy with clinically relevant antibiotics on the multidrug-resistant isolates, by a disk diffusion method and by the checkerboard assay, as well as their capacity to prevent biofilm formation in all four reference strains, by measuring a total biomass using the crystal violet assay.

## 2. Results and Discussion

The structures of (*R*)-6-hydroxymellein (**1**) [9,10], penipurdin A (**2**) [11], acetylquestinol (**3**) [12], tenellic acid C (**5**) [13], and vermixocin A (**8**) [14,15,16] (Figure 1) were elucidated by analysis of their 1D and 2D NMR spectra as well as HRMS data, and also by comparison of their spectral data and signs of optical rotations (Appendix A) with those reported in the literature.

Compounds **3** and **4** were isolated as a 1:3 mixture (estimated by the integration of protons in the ^1^H NMR spectrum). The molecular formula of the minor compound (**3**) was determined as C_18_H_14_O_7_ on the basis of its (+)-HRESIMS *m*/*z* 343.0809 [M + H]^+^ (calculated for C_18_H_15_O_7_, 343.0818), while the molecular formula of the major compound (**4**) was established as C_20_H_18_O_7_ on the basis of (+)-HRESIMS *m*/*z* 371.1124 [M + H]^+^ (calculated for C_20_H_19_O_7_, 371.1131) (Appendix A). The ^1^H and ^13^C signals as well as correlations observed in the COSY, HSQC, and HMBC spectra of a minor component (Appendix A) revealed its identity as acetylquestinol, previously reported from the culture of *Eurotium chevalieri* KUFA0006 [12].

The ^13^C NMR spectrum (Table 1, Appendix A) of the major compound (**4**) showed 20 carbon signals, which, in combination with DEPT and HSQC spectra (Appendix A), can be categorized as two conjugated ketone carbonyls (δ_C_ 186.4 and 183.0), one ester carbonyl (δ_C_ 170.2), three oxygen-bearing sp^2^ (δ_C_ 166.6, 164.1, 162.1), five non-protonated sp^2^ (δ_C_ 146.6, 137.2, 132.6, 115.6, 112.2), four protonated sp^2^ (δ_C_ 125.1, 119.6, 108.3, 105.5), one oxygen-bearing methine sp^3^ (δ_C_ 70.7), one methoxy (δ_C_ 56.7), one methylene sp^2^ (δ_C_ 41.5), and two methyl (δ_C_ 21.4 and 20.0) carbons. The ^1^H NMR spectrum (Appendix A), in conjunction with COSY and HMBC spectra (Table 1, Appendix A), showed the following proton signals: a singlet of a hydrogen-bonded phenolic hydroxyl proton at δ_H_ 13.40, two pairs of *meta*-coupled aromatic protons at δ_H_ 7.15, d (*J* = 1.5 Hz, δ_C_ 125.1)/δ_H_ 7.47, d (*J* = 1.5 Hz, δ_C_ 119.6) and δ_H_ 7.16, d (*J* = 2.2 Hz, δ_C_ 108.3)/δ_H_ 6.78, d (*J* = 2.2 Hz, δ_C_ 105.5), a multiplet at δ_H_ 5.06 (δ_C_ 70.7), a methoxyl singlet at δ_H_ 3.89 (δ_C_ 56.7), a multiplet at δ_H_ 2.92 (δ_C_ 41.5), a methyl singlet at δ_H_ 1.94 (δ_C_ 21.4), and a methyl doublet at δ_H_ 1.20, d (*J* = 6.3 Hz, δ_C_ 20.0). The general feature of the ^1^H and ^13^C NMR spectra of **4** resembled those of **2** [11], except for the presence of the ester carbonyl at δ_C_ 170.2 and a methyl singlet at δ_H_ 1.94 (δ_C_ 21.4), characteristic of an acetyl group. That the substituent on C-3 was 2-acetoxypropyl instead of 2-hydroxylpropyl in **2** was evidenced by the COSY correlations from H_2_-1′ (δ_H_ 2.92, m/δ_C_ 41.5) to H-2′ (δ_H_ 5.06/δ_C_ 70.7), and from H-2′ to H_3_-3′ (δ_H_ 1.20, *d*, *J* = 6.3 Hz/δ_C_ 20.0) as well as by HMBC correlations from H-2′ and a methyl singlet at δ_H_ 1.94 (δ_C_ 21.1) to the carbonyl at δ_C_ 170.2, H-1′ to C-2, C-3 (δ_C_ 146.6), C-4, and from H-2′ to C-3.

The optical rotation of a mixture of **3** and **4** is dextrorotatory, with [α]^20^_D_ + 142.8 (*c* 0.035, MeOH). Since **3** is not a chiral molecule, only **4** is responsible for the optical activity. The structure of **4** corresponds to an acetylated derivative of **2** ([α]^20^_D_ + 35.7 (*c* 0.033, MeOH; Lit. + 33 (*c* 1.14, MeOH)) and is, therefore, named acetylpenipurdin A. Based on the biogenetic consideration, the absolute configuration of C-2′ in **4** should be the same as that of C-2′ in **2**, i.e., (*S*). Additionally, this hypothesis was supported by the same sign of their optical rotations (dextrorotatory), which is based on the report by Singh et al. [17], that 1,3,8-trihydroxy-6-(2′-acetoxyptopyl) anthracene-9,10-dione, isolated from the marine crinoid *Pterometra venusta*, ([α]^25^_D_ + 40, *c* 0.05, MeOH) and its deacetylated product ([α]^25^_D_ + 37, *c* 0.05, MeOH) were both dextrorotatory. Literature search revealed that **4** has never been previously reported.

The (+)-HRESIMS, ^1^H and ^13^C NMR data (Appendix A) of **5** ([α]^20^_D_–7.8 (*c* 0.079, MeOH)) are compatible with those of tenellic acid C ([α]^25^_D_–7.4 (*c* 0.13 g/dL, MeOH)), a biphenyl ether derivative isolated from the aquatic fungus *Dendrospora tenella* [13]; however, the absolute configuration of C-8 was not established. To determine the absolute configuration of C-8, the experimental ECD spectrum of **5** was measured and then compared with a quantum-mechanically simulated spectrum derived from the most significant computational models of (*S*)-**5** (Figure 2, please see the Experimental section for details). Figure 3 shows the visual match between experimental and calculated ECD spectra, with the spectrum of the (8*S*) configuration mostly in phase with the experimental spectrum, while the spectrum of (8*R*) configuration is mostly out of phase, leading to an unambiguous conclusion that the absolute configuration of C-8 is (*S).*

Compound **6** was isolated as an amorphous yellow solid, and its molecular formula C_23_H_28_O_7_ was established based on the (+)-HRESIMS *m*/*z* 417.1915 [M + H]^+^ (calculated for C_23_H_29_O_7_, 417.1913) (Appendix A), requiring 10 degrees of unsaturation. The general feature of the ^1^H and ^13^C NMR spectra of **6** resembled those of **5**. The ^13^C NMR spectrum (Table 2, Appendix A) exhibited 23 carbon signals, which can be categorized, according to DEPT and HSQC spectra (Appendix A), as one aldehyde carbonyl (δ_C_ 189.8), one conjugated carboxy carbonyl (δ_C_ 167.2), four oxygen-bearing sp^2^ (δ_C_ 155.1, 154.8, 150.9, and 142.3), four non-protonated sp^2^ (δ_C_ 136.7, 130.5, 129.8, and 119.3), four protonated sp^2^ (δ_C_ 128.5, 124.2, 118.4, and 110.3), one oxymethine sp^3^ (δ_C_ 73.1), one oxymethylene sp^3^ (δ_C_ 63.9), one methoxy (δ_C_ 62.8), one methylene sp^3^ (δ_C_ 47.1), one methine sp^3^ (δ_C_ 24.9), and four methyl (δ_C_ 23.7, 22.2, 21.0, and 15.7) carbons. The ^1^H NMR spectrum (Appendix A), in combination with the COSY spectrum (Table 2, Appendix A), displayed a singlet of an aldehyde proton at δ_H_ 10.15, the signals of two *ortho*-coupled aromatic protons at δ_H_ 7.25, d (*J* = 8.7 Hz) and 6.32, d (*J* = 8.7 Hz), two *meta*-coupled aromatic protons at δ_H_ 7.14, d (*J* = 2.0 Hz) and 7.11, d (*J* = 2.0 Hz), a double doublet at δ_H_ 4.58 (*J* = 9.2, 3.8 Hz), a methoxy singlet at δ_H_ 3.83, a quartet at δ_H_ 3.25 (*J* = 7.0 Hz), two double of double doublets of two geminally coupled methylene protons at δ_H_ 1.57 (*J* = 13.8, 9.1, 5.0 Hz) and 1.27 (*J* = 13.8, 8.9, 3.9 Hz), a multiplet at δ_H_ 1.71, one methyl singlet at δ_H_ 2.31, one methyl triplet at δ_H_ 1.06 (*J* = 7.0 Hz), and two methyl doublets at δ_H_ 0.88 (*J* = 6.6 Hz) and 0.92 (*J* = 6.6 Hz). The ^13^C and ^1^H chemical shift values and multiplicities of the proton signals revealed that **6** is also a biphenyl ether derivative. Like **5**, one of the benzene ring is 1,2,3,5-tetrasubstituted and another is 1,2,3,4-tetrasubstituted. That the 1,2.3,5-tetrasubstituted benzene ring has a formyl group on C-1′, a hydroxyl group on C-3′, and a methyl group on C-5′ was corroborated by COSY correlations from the doublet at δ_H_ 7.14 (*J* = 2.0 Hz, H-4′/δ_C_ 124.2) to the doublet at δ_H_ 7.11 (*J* = 2.0 Hz, H-6′/δ_C_ 118.4) and the methyl singlet at δ_H_ 2.31 (Me-8′/δ_C_ 21.0), from Me-8′ to H-4′ and H-6′ as well as HMBC correlations (Appendix A) from H-4′ to C-6′, C-8′, C-2′ (δ_C_ 142.3), and C-3′(δ_C_150.9), H-6′ to C-2′, C-7′ (δ_C_ 189.8), and C-8′, Me-8′ to C-4′, C-5′ (δ_C_ 136.7), and C-6′, H-7′ (δ_H_ 10.15, s) to C-1′ (δ_C_ 129.8) C-5′ and C-6′. That another benzene ring has a carboxyl substituent on C-1, a methoxyl group on C-2, and an alkyl sidechain on C-3 was substantiated by COSY correlations from H-4 (δ_H_ 7.25, d, *J* = 8.7 Hz/δ_C_ 128.5) to H-5 (δ_H_ 6.32, d, *J* = 8.7 Hz/δ_C_ 110.3) as well as HMBC correlations from H-4 to C-2 (δ_C_ 155.1), C-6 (δ_C_ 154.8) and C-8 (δ_C_ 73.1), H-5 to C-1(δ_C_ 119.3), C-3 (δ_C_ 130.5), C-6 and C-7 (δ_C_ 167.2), the methoxyl singlet at δ_H_ 3.83 (δ_C_ 62.8) to C-2. That the substituent on C-3 is 1-ethoxy-3-methylbutyl is corroborated by COSY correlations from H-8 (δ_H_ 4.58, dd, *J* = 9.2, 3.8 Hz/δ_C_ 73.1) to H_2_-9 (δ_H_ 1.57, ddd, *J* = 13.8, 9.1, 5.0 Hz and 1.27, ddd, *J* = 13.8, 8.9, 3.9 Hz/δ_C_ 47.1), H_2_-9 to H-10 (δ_H_ 1.71, m/δ_C_ 24.9), H-10 to Me-11 (δ_H_ 0.88, d, *J* = 6.6 Hz/δ_C_ 23.7) and Me-12 (δ_H_ 0.92, d, *J* = 6.6 Hz/δ_C_ 22.2), and H-14 (δ_H_ 3.25, q, *J* = 7.0 Hz/δ_C_ 63.9) to Me-15 (δ_H_ 1.06, t, *J* = 7.0 Hz/δ_C_ 15.7). This was supported by HMBC correlations from H-8 to C-9, C-14 and C-3, H-14 to C-8, Me-15, H-9 to C-8, C-11, Me-11 to C-9, C-10, C-12, and Me-12 to C-9, C-10 and C-11.

Taking together the molecular formula and the partial structures, the two substituted benzene rings must be connected by an ether bridge through C-6 and C-2′, forming a biphenyl ether derivative. The only difference between the structure of **6** and that of **5** is that the acetoxy group on C-8 in **5** was replaced by the ethoxy group in **6**. Since **6** cannot be obtained as a suitable crystal for X-ray analysis, we attempted to determine the absolute configuration of C-8 of **6** by comparing the experimental and calculated ECD spectra. Unfortunately, **6** does not produce an ECD spectrum at a concentration that normally gives a visible spectrum for other compounds of this series. Therefore, based on the biogenic consideration, we presume that the absolute configuration of C-8 in **6** is the same as that in **5**. Moreover, this hypothesis is supported by the fact that both **5** and **6** are levorotatory. Thus, the absolute configuration of C-8 in **6** was proposed as (*S*). Since **6** has not been previously reported, it was named neospinosic acid.

Compound **7** was isolated as a yellow viscous oil, and its molecular formula was established as C_21_H_24_O_6_ on the basis of (+)-HRESIMS *m*/*z* 373.1652 [M + H]^+^ (calculated for C_21_H_25_O_6_, 373.1651) (Appendix A), corresponding to 10 degrees of unsaturation. The ^13^C NMR spectrum (Table 3, Appendix A) displayed 21 carbon signals, which, in combination with DEPT and HSQC spectra (Appendix A), can be categorized as nine non-protonated sp^2^ (δ_C_ 161.6, 160.2, 157.9, 145.9, 143.4, 138.3, 136.3, 135.6, and 114.3), four protonated sp^2^ (δ_C_ 132.8, 125.9, 119.9, and 115.6), one methoxy (δ_C_ 63.1), one oxymethine sp^3^ (δ_C_ 64.4), one methine sp^3^ (δ_C_ 24.8), one oxymethylene sp^3^ (δ_C_ 57.9), one methylene sp^3^ (δ_C_ 48.3), and three methyl (δ_C_ 23.9, 22.1, and 20.9) carbons. The ^1^H NMR spectrum (Appendix A), in combination with the COSY spectrum (Table 3, Appendix A), showed two *ortho*-coupled aromatic protons at δ_H_ 7.67, d (*J* = 8.6 Hz) and 7.21, d (*J* = 8.6 Hz), two *meta*-coupled aromatic protons at δ_H_ 7.12, d (*J* = 0.5 Hz) and 7.11, brs, one triplet at δ_H_ 5.34 (*J* = 5.8 Hz), one double of double doublet at δ_H_ 4.87 (*J* = 9.2, 4.9, 4.2 Hz), two doublet at δ_H_ 4.72 (*J* = 5.7 Hz) and 5.13 (*J* = 4.9 Hz), a methoxy singlet at δ_H_ 3.76, two geminally coupled double of double doublets at δ_H_ 1.24 (*J* = 13.7, 9.2, 4.2 Hz) and δ_H_ 1.44 (*J* = 13.7, 9.2, 4.9 Hz), a multiplet at δ_H_ 1.72, one methyl singlet at δ_H_ 2.28, and two methyl doublets at δ_H_ 0.86 (*J* = 6.7 Hz) and 0.90 (*J* = 6.7 Hz). The ^1^H and ^13^C chemical shift values and multiplicities of the aromatic proton signals suggested the presence of two substituted benzene rings in the molecule. That one of the benzene ring is 1,2,3,4-tetrasubstituted, with a methoxyl group on C-2 and an oxygenated substituent on C-6, was evidenced by the COSY correlations (Table 3) from the doublet at δ_H_ 7.67 (*J* = 8.6 Hz, H-4/δ_C_ 132.8) to another doublet at δ_H_ 7.21 (*J* = 8.6 Hz, H-5/δ_C_ 115.6) as well as by HMBC correlations (Table 3, Appendix A) from H-4 to the carbons at δ_C_ 160.2 (C-6), 157.9 (C-2) and the oxymethine sp^3^ carbon at δ_C_ 64.4, H-5 to the carbons at δ_C_ 114.3 (C-1), 138.3 (C-3), C-6, a methoxyl singlet at δ_H_ 3.76 (δ_C_ 63.1) to C-2. The chemical shift value of C-1 suggested that it was substituted by a carbonyl group. That the substituent on C-3 is 1-hydroxy-3-methylbutyl was substantiated by COSY correlations (Table 3) from H-8 (δ_H_ 4.87, ddd, *J* = 9.2, 4.9, 4.2 Hz/δ_C_ 64.4) to the double of double doublet at δ_H_ 1.44 (*J* = 13.7, 9.2, 4.9 Hz, H-9b/δ_C_ 48.3) and a doublet at δ_H_ 5.13 (*J* = 4.9 Hz, OH-8), and HMBC correlations (Table 3) from H-9b to C-8, OH-8 to C-9, the methyl doublet at δ_H_ 0.86 (*J* = 6.7 Hz/δ_C_ 23.9; Me-11) to the carbon at δ_C_ 48.3 (C-9), 24.8 (C-10), 22.1 (Me-12), the doublet at 0.90 (*J* = 6.7 Hz/δ_C_ 22.1; Me-12) to C-9, C-10 and Me-11.

That another benzene ring is 1,2,3,5-tetrasubstituted, with a methyl substituent on C-5′ and a hydroxymethyl group on C-1′, was substantiated by a COSY correlation (Table 3) from the triplet at δ_H_ 5.34 (*J* = 5.8 Hz, OH-7′) to a doublet at δ_H_ 4.72 (*J* = 5.8 Hz, H_2_-7′/δ_C_ 57.9) and HMBC correlations (Table 3) from H_2_-7′ to the carbons at δ_C_ 125.9 (C-6′), 135.6 (C-1′), 145.9 (C-2′), the methyl singlet at δ_H_ 2.28 (Me-8′/δ_C_ 20.9) to the carbons at δ_C_ 119.9 (C-4′), 136.3 (C-5′), and C-6′, a doublet at δ_H_ 7.12 (*J* = 0.5 Hz, H-6′/δ_C_ 125.9) to C-4′, C-2′ (δ_C_ 145.9), C-7′ and Me-8′, and from a broad singlet at δ_H_ 7.11 (H-4′/δ_C_ 119.9) to C-2′, C-3′ (δ_C_ 143.4) and C-6′. The chemical shift values of C-2′ and C-3′ suggested that they are oxygen-bearing aromatic carbons.

Considering the ^1^H and ^13^C chemical shift values and the molecular formula, the two substituted benzene rings must be connected by an ether bridge between C-2′ and C-6 as well as between the oxygen atom on C-3′ and the carbonyl on C-1, thus forming a 5*H*-1,4-dioxepin-5-one ring. Therefore, the carbon at δ_C_ 161.6 was assigned to C-7.

Since **7** has one stereogenic carbon (C-8), it is necessary to determine its absolute configuration. Compound **7** was isolated as a viscous oil, which was not able to determine the absolute configuration of C-8 by X-ray crystallography. Therefore, the absolute configuration of C-8 in **7** was determined by ECD. For this effect, the experimental ECD spectrum of **7** was measured and then compared with a quantum-mechanically simulated spectrum derived from the most significant computational models of (*S*)-**7** (Figure 4; please see Experimental section for details). Figure 5 shows a good match between experimental and calculated ECD spectra, with the two spectra in phase, leading to a conclusion that the absolute configuration of C-8 is (*S*).

Literature search through SciFinder revealed that **7** has never been previously described, and, therefore, was named spinolactone.

Interestingly, Nishida et al. [18] reported the structure of a similar compound containing a 11*H*-dibenzo[b,e][1,4]dioxepin-11-one scaffold, named purpactin C’ which was obtained by conversion of purpactin C, isolated from a fermentation broth of *Penicillium purpurogenum* FO-608. However, the authors only presented its HREI-MS, ^1^H and ^13^C NMR data (CDCl_3_, 300 and 75 MHz) of purpactin C’. Later on, Chen et al. isolated purpactin C’ from a gorgonian-derived *Talaromyces* sp. [19], whereas Daengrot et al. also described the isolation of the same compound from a soil-derived *Penicillium aculeatum* PSU-RSPG105 [16]. In both cases, the authors reported neither its NMR data nor absolute configuration of the stereogenic carbon in the side chain but only referred to the work of Nishida et al. [18].

Since the two benzene rings of **5**–**8** possess the same substitution patterns, it is clearly that they share the same biosynthetic origin and route. Condensation of acetyl CoA (**I**) and malonyl CoA (**II**) gives an octaketide **III**, which undergoes a cyclization to give an intermediate **IV**. However, instead of enolization, one of the ketone carbonyl in ring C undergoes a reduction to form a secondary alcohol, which, after oxidation of the methylene group in ring B, gives rise to **VI**. Decarboxylation of ring A and dehydration of the secondary alcohol in ring C of **VI** gives rise to the anthraquinone **VII**. Prenylation on the activated carbon in **VII** by dimethylallyl pyrophosphate (DMAPP) gives rise to a prenyl intermediate **VIII**, followed by enolization to give an intermediate **IX**. Methylation of the phenolic hydroxyl group, *ortho* to the prenyl group, by SAM, leads to the formation of **X**. Oxidative cleavage of the anthraquinone ring gives rise to **XI**. Rotation of the bond between the benzene ring (A) and the carbonyl group in **XI** to **XII** allows a nucleophilic substitution of the hydroxyl group to give an intermediate **XIII**. Oxidation of the aldehyde to a carboxylic acid and oxidation of the double bond of the prenyl side chain lead to an intermediate **XIV**, which, after dehydration, gives **XV**. Regiospecific hydration of the double bond of the side chain of **XV** gives **XVI**, which, after reduction of one of the carboxylic acid group to aldehyde, results in a formation of **XVII**. Acetylation of the hydroxyl group of the side chain leads to the formation of **5**, which, after reduction of the carbonyl carbon of the acetyl group, gives rise to **6** (Figure 6).

Reduction of the aldehyde group on ring A of **XVII** to a primary alcohol in **XVIII** or **XIX**, followed by esterification of the carboxyl group by the phenolic hydroxyl group (in **XIX**) or by a hydroxyl group of the primary alcohol (in **XVIII**) leads to the formation of **7** and **8**, respectively (Figure 7).

The antimicrobial activity of **2** and **5**–**8** was evaluated against several reference bacterial species and multidrug-resistant isolates (Table 4); however, only **7** exhibited antibacterial activity against *Enterococcus faecalis* B3/101 with a MIC value of 64 µg/mL (Table 4). The MBC was more than one-fold higher than the MIC, suggesting a bacteriostatic effect.

Although **5** and **6** did not exhibit antibacterial activity, they were able to significantly inhibit biofilm formation in three of the four reference strains used in this study (Table 5): *Escherichia coli* ATCC 25922 (both **5** and **6**), *Staphylococcus aureus* ATCC 29213 (both **5** and **6**), and *E. faecalis* ATCC 29212 (**5**). A more extensive effect was found for **6**, which displayed the strongest inhibitory activity (56.00 ± 0.06) (mean ± SD) in *S. aureus* ATCC 29213.

This result led us to investigate the influence of **6** in both biofilm viability (Figure 8) and its matrix spatial arrangement (Figure 9). After 8 h of incubation, the viability of the biofilm of *S. aureus* ATCC 29213 was significantly affected by **6**, exhibiting a percentage of control of 1.80 ± 0.0014, representing a viability reduction of 98%. On the contrary, after 24 h of incubation, the percentage of control increased to 89.65 ± 0.0021, showing only a 10% viability reduction. Although the results herein presented suggest a sublethal effect of **6** on a specific molecular or structural target of *S. aureus*, that could be reversed over time due to genetic and phenotypic adaptability of bacteria; however, it cannot be ruled out that the compound may undergo some degradation or biodegradation. Further studies are warranted to shed more light on this subject.

Regarding the effect on biofilm extracellular polymeric substances, **6** caused an increased number of channels, homogeneously distributed by the biofilm, after 8 h of incubation (Figure 9). However, after 24 h of incubation, this biofilm did not maintain its structure, appearing quite similar to the control (data not shown). In fact, *S. aureus* ATCC 29213 typically produces a dense biofilm structure with lower number of observed channels. Biofilm interstitial voids (channels) are physiologically relevant for diffusion and circulation of nutrients, oxygen and essential substances. Factors such as cell-to-cell communication and alterations in attachment of bacterial cell to surfaces can influence the dynamic of biofilm, namely the evolution of the channels. Formation of channels was shown to be affected by molecules like surfactants, which have the ability to modulate gene expression and to maintain open channels [20,21]. Nonetheless, the present study highlights the promising results of **6** in *S. aureus* biofilm early development. Understanding the antibiofilm dynamic in the presence of **6** and its stability is essential to evaluate its activity, especially within the first 8 h of incubation. Compound **5** was also investigated for its potential synergy with clinically relevant antibiotics on the multidrug-resistant isolates, by both disk diffusion method and checkerboard assay; however, no interactions were observed.

Compounds **2** and **5**–**8** were also tested for their in vitro acetylcholinesterase (AChE) inhibitory activity by a modified Ellman’s method [22]; however, none of the tested compounds showed inhibition of the enzyme at concentrations as high as 80 µM (a positive control galantamine showed a percentage inhibition of 94.82% at 80 µM, and an IC_50_ value of 16.76 μM). Additionally, **2** and **5**–**8** were also evaluated for their anti-tyrosinase activity at the maximum concentration of 200 µM by using a modified microplate assay as described previously [23]. All the tested compounds, except **6**, inhibited tyrosinase activity. However, as **8** showed a percentage of inhibition higher than 50% at 200 µM, its IC_50_ value (177.03 ± 8.17 µM) was obtained at lower doses (i.e., 150 and 100 µM), indicating its moderate anti-tyrosinase activity. On the contrary, **2**, **5**, **7** showed weak inhibitory effects. Table 6 shows percent inhibition at 200 µM and IC_50_ values (µM) of the tested compounds.

## 3. Experimental Section

### 3.1. General Experimental Procedures

The melting points were determined on a Stuart Melting Point Apparatus SMP3 (Bibby Sterilin, Stone, Staffordshire, UK) and are uncorrected. Optical rotations were measured on an ADP410 Polarimeter (Bellingham + Stanley Ltd., Tunbridge Wells, Kent, UK). ^1^H and ^13^C NMR spectra were recorded at ambient temperature on a Bruker AMC instrument (Bruker Biosciences Corporation, Billerica, MA, USA) operating at 300 or 500 and 75 or 125 MHz, respectively. High-resolution mass spectra were measured with a Waters Xevo QToF mass spectrometer (Waters Corporations, Milford, MA, USA) coupled to a Waters Aquity UPLC system. A Merck (Darmstadt, Germany) silica gel GF_254_ was used for preparative TLC, and Merck Si gel 60 (0.2–0.5 mm), Li Chroprep silica gel and Sephadex LH 20 were used for column chromatography.

### 3.2. Fungal Material

The fungus was isolated from a marine sponge *Mycale* sp., which was collected by scuba diving at a depth of 10–15 m from the coral reef at Samae San Island (12°34′36.64” N 100°56′59.69” E), Chonburi province, Thailand, in September 2016. The sponge was washed with sterilized seawater three times, and then dried on a sterile filter paper under sterile aseptic condition. The sponge was cut into small pieces (5 × 5 mm), and four pieces were placed on Petri dish plates containing 15 mL potato dextrose agar (PDA) medium mixed with 300 mg/L of streptomycin sulfate and incubated at room temperature for 7 days. The hyphal tips emerging from sponge pieces were individually transferred onto PDA slant.

The fungal strain KUFA 1047 was identified as *Neosartorya spinosa*, based on morphological characteristics. This identification was confirmed by molecular techniques using internal transcribed spacer (ITS) primers. DNA was extracted from young mycelia following a modified Murray and Thompson method [24]. The universal primer pairs ITS1 and ITS4 were used for ITS gene amplification [25]. PCR reactions were conducted on a thermal cycler and DNA sequencing analyses were performed using the dideoxyribonucleotide chain termination method [26] by Macrogen Inc. (Seoul, South Korea). The DNA sequences were edited using FinchTV software and submitted into the BLAST program for alignment and compared with that of fungal species in the NCBI database (http://www.ncbi.nlm.nih.gov/, accessed on 15 January 2017). Its gene sequences were deposited in GenBank with an accession number MT814287. The pure cultures were maintained at Kasetsart University Fungal Collection, Department of Plant Pathology, Faculty of Agriculture, Kasetsart University, Bangkok, Thailand.

### 3.3. Extraction and Isolation

The mycelium plugs of *Neosartorya spinosa* KUFA 1047 were transferred into 500 mL Erlenmeyer flasks containing 200 mL of potato dextrose broth (PDB) and incubated on a rotary shaker at 120 rpm for 7 days at room temperature for preparing a mycelial suspension. Fifty 1 L Erlenmeyer flasks, each containing 300 g sterile cooked rice and then inoculated with 20 mL of mycelial suspension in each flask and incubated at room temperature for 30 days. Then, 500 mL of EtOAc were added to each flask and macerated for 7 days and then filtered with Whatman filter paper N^o^.1 (GE Healthcare UK Limited, Buckinghamshire, UK). The organic solutions were combined and then evaporated under reduced pressure to give 227.8 g of crude EtOAc extracts of *N. spinosa* KUFA 1047. The crude EtOAc extract of *N. spinosa* KUFA 1047 was dissolved in 500 mL of EtOAc and then washed with a 5% NaHCO_3_ solution (4 × 250 mL). The resulting organic phase was washed with deionized water (3 × 500 mL) and dried with anhydrous Na_2_SO_4_, filtered, and evaporated under reduced pressure, to obtain 31.2 g of the crude EtOAc extract, which was applied on a column chromatography of silica gel 60 (350 g) and eluted with mixtures of petrol-CHCl_3_ and CHCl_3_-Me_2_CO, wherein 250 mL fractions (frs) were collected as follows: frs 1–192 (petrol-CHCl_3_, 3:7), 193–282 (petrol–CHCl_3_, 1:9), 283–432 (CHCl_3_–Me_2_CO, 9:1), and 433–598 (CHCl_3_–Me_2_CO, 7:3). Frs 109–155 were combined (980 mg) and applied over a column chromatography of Sephadex LH-20 (15 g) and eluted with MeOH, wherein 35 sub-fractions (sfrs) of 1 mL were collected. Sfrs 9–22 were combined (861.5 mg) and applied over a column chromatography of Sephadex LH-20 (15 g) and eluted with CHCl_3_, wherein 22 ssfrs of 1 mL were collected. Ssfrs 8–13 were combined (483 mg) and applied over a column chromatography of Sephadex LH-20 (5 g) and eluted with MeOH, wherein 14 frs of 0.25 mL were collected. Frs 5–9 were combined (237.4 mg) and applied over a column chromatography of Sephadex LH-20 (5 g) and eluted with MeOH, wherein 12 sfrs of 0.25 mL were collected. Sfrs 9–12 were combined and to give 36.9 mg of yellow viscous mass of **8**. Frs 156–200 were combined (180.0 mg) and applied over a column chromatography of Sephadex LH-20 (5 g), and eluted with MeOH, wherein 17 sfrs of 1 mL were collected. Sfrs 8-17 were combined (42.8 mg) and applied over a column chromatography of Sephadex LH–20 (5 g), and eluted with MeOH, wherein 19 ssfrs of 0.5 mL were collected. Ssfrs 14-19 were combined and to give 5.4 mg of a mixture of **3** and **4**. Sfrs 5–7 (75.5 mg) from the first Sephadex LH-20 column were combined with ssfrs 3-13 (36.9) of the second Sephadex LH-20 column (112.4 mg) and applied over a column chromatography of Sephadex LH-20 (5 g), and eluted with CHCl_3_, wherein 22 frs of 1 mL were collected. Frs 8–22 were combined (83.6 mg) and purified by TLC (silica gel 60 G_254_, CHCl_3_: Me_2_CO, 9:1) to give 8.8 mg of **7**. Frs 288–291 were combined (226.4 mg) and precipitated in CHCl_3_ to give 10.3 mg **2**. The mother liquor (209.8 mg) was applied over a column chromatography of Sephadex LH-20 (5 g) and eluted with MeOH, wherein 29 sfrs of 0.25 mL were collected. Sfrs 8–15 were combined (128.4 mg) and applied over a column chromatography of Sephadex LH-20 (5 g) and eluted with CHCl_3_, wherein 23 ssfrs of 1 mL were collected. Ssfrs 19–23 were combined (43.2 mg) and purified by TLC (silica gel 60 G_254_, petrol: CHCl_3_: Me_2_CO, 2: 85:13) to give 14.3 mg of **6**. Frs 293–298 were combined (798.7 mg) and precipitated in CHCl_3_ to give additional 9.4 mg of **2**. Frs 339–379 were combined (506.6 mg) and purified by TLC (silica gel 60 G_254_, CHCl_3_: Me_2_CO, 85:15). One of the bands from the TLC plates (128.4 mg) was applied over a column chromatography of Sephadex LH-20 (5 g) and eluted with MeOH, wherein 19 sfrs of 0.25 mL were collected. Sfrs 4–19 were combined (108.1 mg) and precipitated in CHCl_3_ to give 39.3 mg of **5**. Frs 299–338 (1.29 g), the mother liquor of frs 293–298 (775.9 mg), and the rest of the TLC bands of frs 339–379 (65.1 mg) were combined (2.13 mg) and applied over a column chromatography of Li Chroprep^®^ silica gel (35 g) and eluted with mixtures of petrol–CHCl_3_ and CHCl_3_–Me_2_CO, wherein 112 sfrs of 100 mL were collected as follow: 1–26 (petrol–CHCl_3_, 3:7), 27–75 (petrol–CHCl_3_, 1:9), 76–99 (CHCl_3_–Me_2_CO, 9:1), and 100–112 (CHCl_3_–Me_2_CO, 7:3). Sfrs 7–8 were combined (41.5 mg) and applied over a column chromatography of Li Chroprep^®^ silica gel (8 g) and eluted with mixtures of petrol–CHCl_3_ and CHCl_3_–Me_2_CO, wherein 33 ssfrs of 25 mL were collected as follow: ssfrs 1–4 (petrol–CHCl_3_, 1:1), 5–17 (petrol–CHCl_3_, 3:7), 18–25 (petrol–CHCl_3_, 1:9), 26–30 (CHCl_3_–Me_2_CO, 9:1), and 31–33 (CHCl_3_–Me_2_CO, 7:3). Ssfrs 9–11 were combined (7.6 mg) and precipitated in CHCl_3_ to give 6.6 mg of **1**.

#### 3.3.1. Acetylpenipurdin A (**4**)

For ^1^H and ^13^C NMR spectroscopic data (CDCl_3_, 300 and 75 MHz), see Table 1. (+)-HRESIMS *m/z* 371.1124 (M + H)^+^ (calcd for C_20_H_19_O_7_, 329.1131).

#### 3.3.2. Neospinosic Acid (**6**)

Yellow amorphous solid. [α]_D_^20^ -166.7 (*c* = 0.042, MeOH). IR (KBr) υ_max_ 3447 (OH), 2956, 2930, 2869, 1683 (CHO), 1653, 1616, 1596, 1472, 1323, and 12792 cm^-1^. For ^1^H and ^13^C spectroscopic data (DMSO-*d6*, 300 and 75 MHz), see Table 2; (+)-HRESIMS *m/z* 417.1915 (M + H)^+^ (calcd for C_23_H_29_O_7_, 417.1913); 439.1735 (M + Na)^+^ (calcd for C_23_H_28_O_7_Na, 439.1733).

#### 3.3.3. Spinolactone (**7**)

Yellow viscous oil. [α]_D_^20^ -311.5 (*c* = 0.06, MeOH). IR (KBr) ν_max_ 3423 (OH), 2955, 2927, 2868, 1746, 1596, 1521, 1488, 1470, 1225, and 1202 cm^−1^. For ^1^H and ^13^C spectroscopic data (DMSO-*d*6, 500 and 125 MHz), see Table 3; (+)-HRESIMS *m/z* 373.1652 (M+H)^+^ (calcd for C_21_H_25_O_6_, 373.1651).

### 3.4. Electronic Circular Dichroism (ECD)

The experimental UV and ECD spectra of **5** and **7** (*ca.* 1 mg/mL in methanol and acetonitrile) were obtained in a Jasco J-815 CD spectropolarimeter (Jasco Europe S.R.L., Cremella, Italy) with a 0.1 mm cuvette and 12 accumulations. The simulated ECD spectrum was obtained by first determining all the relevant conformers of the (*S*)-**5** computational model. Its conformational space was developed by rotating by 90, 120, or 180 degrees all the single, non-cyclic, bonds, depending on the MM2 torsion energy minima. The large number of conformational degrees of freedom of **5** resulted in a huge number of conformers, which were MM2 minimized with PerkinElmer’s Chem3D Ultra 20.1.0.110 (PerkinElmer Inc., Waltham, MA, USA) and ChemScript (PerkinElmer Inc., Waltham, MA, USA) and filtered to eliminate like conformations with VeraChem’s Vconf 2.0 (VeraChem LLC, Germantown, MA, USA), resulting in 351 different conformers. Since the MM2 energies of all of these conformers fell within an interval of 3 kcal/mol, many with very similar values, all the 351 conformers were minimized using the semi-empirical method PM6/methanol using Gaussian 16W (Gaussian Inc., Wallingford, NY, USA) and then filtered again. Out of these, 351 PM6 conformers, 20 were found within a 2 kcal/mol interval (about 85% of the population) and were subjected to a final minimization round using the quantum mechanical DFT method B3LYP/6-31G/methanol method (Gaussian 16W), which was also used to calculate its first 70 ECD transitions (TDDFT). The line spectrum for each one of the 20 conformations was built by applying a Gaussian line broadening of 0.3 eV to each computed transition with a constant UV shift of −4 nm. The final ECD spectrum was obtained by the Boltzmann-weighted sum of the 20 line spectra [27].

The simulated ECD spectrum was obtained by first determining all the relevant conformers of the (*S*)-**7** computational model. Its conformational space was developed by rotating by 45 degrees all the single, non-cyclic, bonds for each of the two possible bends about the 7-cycle oxygens. The resulting conformers (over 1000) were MM2 minimized (with PerkinElmer’s Chem3D Ultra) and filtered to eliminate like conformations (with VeraChem’s Vconf 2.0). The lowest 104 conformers, representing about 99% of MM2 total conformer energy, were further minimized using the PM6/acetonitrile semi-empirical method (Gaussian Inc.’s Gaussian 16W) and filtered. The lowest 96% PM6 energy conformers (21 models) were subjected to a final minimization round using the quantum mechanical DFT method B3LYP/6-31G/acetonitrile method (Gaussian 16W), which was also used to calculate its first 70 ECD transitions (TDDFT). The line spectrum for each one of the 21 conformations was built by applying a Gaussian line broadening of 0.2 eV to each computed transition with a constant UV shift of –5 nm. The final ECD spectrum was obtained by the Boltzmann-weighted sum of the 21 line spectra [27].

### 3.5. Antibacterial Activity Bioassays

#### 3.5.1. Bacterial Strains and Growth Conditions

Four reference strains obtained from the American Type Culture Collection were included in this study: two Gram-positive (*Staphylococcus aureus* ATCC 29213 and *Enterococcus faecalis* ATCC 29212), two Gram-negative (*Escherichia coli* ATCC 25922 and *Pseudomonas aeruginosa* ATCC 27853), and one clinical isolate (*E. coli* SA/2, an extended-spectrum β-lactamase producer-ESBL) and two environmental isolates: *S. aureus* 74/24 [28], a methicillin-resistant isolate (MRSA), and *E. faecalis* B3/101 [29] a vancomycin-resistant (VRE) isolate. All bacterial strains were cultured in MH agar (MH-BioKar Diagnostics, Allone, France) and incubated overnight at 37 °C before each assay, in order to obtain fresh cultures. Stock solutions of the compounds were prepared in dimethyl sulfoxide (DMSO—Alfa Aesar, Kandel, Germany), kept at −20 °C, and freshly diluted in the appropriate culture media before each assay. All stock solutions were prepared at final concentration of 10 mg/mL and, in all experiments, in-test concentrations of DMSO were kept below 1%, as recommended by the Clinical and Laboratory Standards Institute [30].

#### 3.5.2. Antimicrobial Susceptibility Testing

The Kirby–Bauer method was used to screen the antimicrobial activity of the compounds according to CLSI recommendations [31]. Briefly, sterile blank paper disks with 6 mm diameter (Liofilchem, Roseto degli Abruzzi, TE, Italy) were impregnated with 15 µg of each compound and placed on MH plates previously inoculated with a bacterial inoculum equal to 0.5 McFarland turbidity. Blank paper disks impregnated with DMSO were used as negative control. MH inoculated plates were incubated for 18–20 h at 37 °C and afterwards the diameter of the inhibition zones was measured in mm.

Minimal inhibitory concentrations (MIC) were determined by the broth microdilution method, as recommended by the CLSI [32]. Two-fold serial dilutions of the compounds were prepared in cation-adjusted Mueller–Hinton broth (CAMHB- Sigma-Aldrich, St. Louis, MO, USA). The tested concentrations ranged from 1 to 64 µg/mL, in order to keep in-test concentrations of DMSO below 1%, avoiding bacterial growth inhibition. Colony-forming unit counts of the inoculum were conducted to ensure that the final inoculum size closely approximated the intended number (5 × 10^5^ CFU/mL). The 96-well U-shaped untreated polystyrene plates were incubated for 16–20 h at 37 °C and the MIC was determined as the lowest concentration of compound that prevented visible growth. During the essays, ceftazidime hidrate (CAZ, Sigma-Aldrich, St. Louis, MO, USA) and kanamycin monosulfate (KAN, Duchefa Biochemie, Haarlem, The Netherlands) were used as positive control of *S. aureus* ATCC 29213 and *E. faecalis* ATCC 29212, respectively. The minimal bactericidal concentration (MBC) was determined by spreading 10 µL of the content of the wells with no visible growth on MH plates. The MBC was determined as the lowest concentration of compound at which no colonies grew after overnight incubation at 37 °C [33]. At least three independent assays were conducted for reference and multidrug-resistant strains.

#### 3.5.3. Antibiotic Synergy Testing

To evaluate the combined effect of the compounds tested with clinically relevant antibacterial drugs, the Kirby–Bauer method was used, as previously described [34]. A set of antibiotic disks (Oxoid, Basingstoke, UK), to which the isolates were resistant, was selected: cefotaxime (CTX, 30 µg) for *E. coli* SA/2, vancomycin (VAN, 30 µg) for *E. faecalis* B3/101, and oxacillin (OXA, 1 µg) for *S. aureus* 66/1. Antibiotic disks impregnated with 15 µg of each compound were placed on seeded MH plates. The controls used included antibiotic disks alone, blank paper disks impregnated with 15 µg of each compound alone, and blank disks impregnated with DMSO. Plates with CTX were incubated for 18–20 h and plates with VAN and OXA were incubated for 24 h at 37 °C [30]. Potential synergy was considered when the inhibition halo of an antibiotic disk impregnated with compound was greater than the inhibition halo of the antibiotic or compound-impregnated blank disk alone.

The combined effect of the compounds and clinically relevant antimicrobial drugs was also evaluated by determining the antibiotic MIC in the presence of each compound. Briefly, when it was not possible to determine an MIC value for the test compound, the MIC of CTX (Duchefa Biochemie, Haarlem, The Netherlands), VAN (Oxoid, Basingstoke, England), and OXA (Sigma-Aldrich, St. Louis, MO, USA) for the respective multidrug-resistant strain was determined in the presence of the highest concentration of each compound tested in previous assays (64 µg/mL). The antibiotic tested was serially diluted, whereas the concentration of each compound was kept fixed. Antibiotic MICs were determined as described above. Potential synergy was considered when the antibiotic MIC was lower in the presence of compound [35]. Fractional inhibitory concentrations (FIC) were calculated as follows: FIC of compound = MIC of compound combined with antibiotic/MIC compound alone, and FIC antibiotic = MIC of antibiotic combined with compound/MIC of antibiotic alone. The FIC index (FICI) was calculated as the sum of each FIC and interpreted as follows: FICI ≤ 0.5, “synergy”; 0.5 < FICI ≤ 4, “no interaction”; 4 < FICI, “antagonism” [36].

#### 3.5.4. Biofilm Formation Inhibition Assay

The antibiofilm activity of compounds was evaluated through quantification of total biomass, using the crystal violet method, as previously described [34,37]. Briefly, the highest concentration of compound tested in the MIC assay was added to bacterial suspensions of 1 × 10^6^ CFU/mL prepared in unsupplemented Tryptone Soy broth (TSB, Biokar Diagnostics, Allone, Beauvais, France) or TSB supplemented with 1% (*p*/*v*) glucose (D-(+)-glucose anhydrous for molecular biology, PanReac AppliChem, Barcelona, Spain) for Gram-positive strains. When it was possible to determine a MIC, concentrations ranging from 2 × MIC to ¼ MIC were tested, while keeping in-test concentrations of DMSO below 1%. When it was not possible to determine a MIC, the concentration tested was 64 µg/mL. Controls with appropriate concentration of DMSO, as well as a negative control (TSB or TSB+1% glucose alone), were included. Sterile 96-well flat-bottomed untreated polystyrene microtiter plates were used. After a 24 h incubation at 37 °C, the biofilms were heat-fixed for 1 h at 60 °C and stained with 0.5% (*v*/*v*) crystal violet (Química Clínica Aplicada, Amposta, Spain) for 5 min. The stain was resolubilized with 33% (*v*/*v*) acetic acid (acetic acid 100%, AppliChem, Darmstadt, Germany) and the biofilm biomass was quantified by measuring the absorbance of each sample at 570 nm in a microplate reader (Thermo Scientific Multiskan^®^ FC, Thermo Fisher Scientific, Waltham, MA, USA). The background absorbance (TSB or TSB+1% glucose without inoculum) was subtracted from the absorbance of each sample and the data are presented as percentage of control. Three independent assays were performed for reference strains, with triplicates for each experimental condition.

#### 3.5.5. Biofilm Viability Assay

Considering the antibiofilm potential of **6**, the metabolic activity of *S. aureus* ATCC 29213 biofilm in presence of **6** at a concentration of 64 µg/mL was assessed using MTT (3-(4,5-dimethylthiazol-2-yl)-2,5-diphenyltetrazolium bromide) assay, as described previously [38,39]. Static biofilm was grown by inoculating 1 × 10^6^ CFU/mL bacteria in sterile 96-well flat-bottomed untreated polystyrene microtiter plates containing TSB supplemented with 1% (*w*/*v*) glucose with a positive and a negative controls. After 8 h and 24 h of incubation at 37 °C, non-adherent cells were removed and 100 µL of MTT (5 mg/mL) (Thiazolyl Blue tetrazolium bromide 98%, Alfa Aesar, Kandel, Germany) was added to each well for 2 h at 37 °C. Thereafter, a solubilization solution (16% (*w*/*v*) of sodium dodecyl sulfate (SDS for molecular biology, PanReac AppliChem, Barcelona, Spain) and 50% DMSO (*v*/*v*) were added to dissolve the formazan product into a colored solution. After overnight dissolution at room temperature, biofilm viability was estimated by measuring absorbance of each sample at 570 nm in a microplate reader (Thermo Scientific Multiskan^®^ FC, Thermo Fisher Scientific, Waltham, MA, USA). Biofilm viability was expressed as percentage of control and at least two different experiments were performed in triplicate.

#### 3.5.6. Biofilm Matrix Visualization

To visualize the extracellular polymeric substances (EPS) matrix of the biofilms, rhodamine-labeled concanavalin A (rhodamine-conA) (Vector Laboratories, Burlingame, CA, USA), which specifically binds to D-(+)-glucose and D-(+)-mannose residues on exopolysaccharide (EPS), was used, as previously described [40]. Interaction between **6** at a concentration of 64 µg/mL and the biofilm of *S. aureus* ATCC 29213 was selected for rhodamine-conA staining, as a consequence of its antibiofilm potential. Briefly, bacterial suspensions of 1 × 10^6^ CFU/mL prepared in TSB supplemented with 1% (*w*/*v*) glucose was added to a sterile well chamber (Ibidi, Gräfelfing, Germany). After 8 h of incubation, non-adherent cells were removed from each well and washed with 200 µL of PBS. Then, 100 µL of a rhodamine-conA (10 mg/mL) solution was added to the biofilm for 30 min in the dark at room temperature. Thereafter, the biofilm was washed with 200 µL of PBS and microscopic visualization, using an excitation of 514 nm and an emission wavelength of 600 ± 50 nm.

### 3.6. Acetylcholinesterase Inhibitory Activity Assay

AChE inhibitory assay was performed according to the Ellman’s method [22]. Briefly, 20 µL of 0.22 U/mL AChE in tris buffer (50 mM, pH 8.0) from *Electrophorus electricus* (EC 3.1.1.7, Sigma-Aldrich, St. Louis, MO, USA) was added to the wells containing 10 µL of tested compounds (80 µM in DMSO), 100 µL of 3 mM of 5,5′-dithio-bis-(2-nitrobenzoic acid) (DTNB, Sigma-Aldrich, St. Louis, MO, USA) (in 50 mM tris buffer, pH 8.0), 20 µL of 15 mM acetylthiocholine iodide (Sigma-Aldrich, St. Louis, MO, USA) (in 50 mM tris buffer, pH 8.0), and 100 µL of 50 mM tris buffer (pH 8.0). Absorbance of the colored-end product was measured at 412 nm for 5 min, with 30 s intervals (BioTek Synergy™ HT Microplate Reader, Winooski, VT, USA). Controls containing 10 µL of DMSO instead of the tested compounds and reaction blanks containing 20 µL of buffer (0.1% (*w*/*v*) bovine serum albumin in 50 mM Tris-HCl) instead of the enzyme and 10 μL of DMSO instead of the tested compounds were made. In this assay, sample blanks containing 20 µL of buffer (0.1% (*w*/*v*) bovine serum albumin in 50 mM Tris-HCl) instead of AChE were also performed. The percentage of enzymatic inhibition was calculated as:Percentage inhibition = 100 − [(*S* − *So*)/(*C* − *Co*)] × 100
where *C* is the absorbance of the control, *Co* is the absorbance of reaction blank, *S* is the absorbance in the presence of the tested compounds, and *So* is the absorbance of sample blanks. All experiments were done in triplicate and galantamine (Sigma-Aldrich, St. Louis, MO, USA), tested at concentrations of 80, 10, 5, and 3.6 µM in DMSO, was used as a positive control as well as for validating the method. The inhibitory activities of the tested compounds toward AChE were expressed as percentage of inhibition as indicated previously. The IC_50_ value of galantamine was obtained by interpolation from a linear regression analysis.

### 3.7. Tyrosinase Inhibitory Activity Assay

Tyrosinase inhibitory assay was performed according to the method previously described [23]. Briefly, 20 μL of the mushroom tyrosinase (EC 1.14.18.1, Sigma-Aldrich, St. Louis, MO, USA, 480 U/mL) in 20 mM phosphate buffer was added to the wells containing 20 μL of the tested compounds (200 μM in DMSO), and 140 µL of 20 mM phosphate buffer (pH 6.8). After incubation at 25 °C for 10 min, 20 μL of 0.85 mM L-DOPA (Sigma-Aldrich, St. Louis, MO, USA) in phosphate buffer (pH 6.8) was added and the absorbance of the colored-end product was measured at 25 °C, 11 times for 10 min., with 1 min. intervals at 475 nm (BioTek SynergyTM HT Microplate Reader, Winooski, VT USA). Controls containing 20 μL of DMSO instead of the tested compounds, and reaction blanks containing 20 µL of 20 mM phosphate buffer (pH 6.8) instead of tyrosinase, and 20 μL of DMSO instead of the tested compounds were performed. Moreover, sample blanks containing 20 µL of 20 mM phosphate buffer (pH 6.8) instead of tyrosinase were made. The percentage inhibition of tyrosinase activity was calculated as:Percentage inhibition = 100 × [1 − (*S* − *So*)/(*C* − *Co*)]
where *S* is the absorbance in presence of the tested compounds, *So* is the absorbance of sample blanks, *C* is the absorbance of the control, and *Co* is the absorbance of reaction blank. All experiments are done in triplicate and kojic acid (Sigma-Aldrich, Saint Louis, MO, USA) at concentrations of 200, 100, 25, 12.5, 8 and 5 µM was used as a positive control. The inhibitory activities of the compounds towards tyrosinase were expressed as per-centage of inhibition as indicated previously. The IC50 value of kojic acid was obtained by interpolation from a linear regression analysis.

### 3.8. Statistical Analysis

Data were reported as means ± standard error of the mean (SEM) of at least three independent experiments. Statistical analysis of the results was performed with GraphPad Prism (GraphPad Software, San Diego, CA, USA). Unpaired *t*-test was carried out to test for any significant differences between the means. Differences at the 5% confidence level were considered significant.

## 4. Conclusions

The EtOAc extract from a solid culture of a marine-derived fungus *Neosartorya spinosa* KUFA1047, isolated from a marine sponge *Mycale* sp. collected in the Gulf of Thailand, furnished five previously reported secondary metabolites *viz*. (*R*)-6-hydroxymellein (**1**), penipurdin A (**2**), acetylquestinol (**3**), tenellic acid C (**5**) and vermixocin A (**8**), in addition to three previously unreported compounds, including acetylpenipurdin A (**4**), neospinosic acid (**6**), and spinolactone (**7**). All the isolated compounds, except **1**, were assayed for in vitro anticholinesterase and anti-tyrosinase activities. Although none of the test compounds exhibited anticholinesterase activity, **2**, **5**, and **7** exhibited weak anti-tyrosinase activity, while **8** showed moderate inhibitory activity against a mushroom tyrosinase. Compounds **2** and **5**–**8** were also assayed for their antibacterial activity against several reference bacterial species and multidrug-resistant isolates; however, only **7** exhibited antibacterial activity against *Enterococcus faecalis* B3/101 with a MIC value of 64 µg/mL. Since the MBC was more than one-fold higher than the MIC, **7** was suggested to exert a bacteriostatic effect. Interestingly, although **5** and **6** did not exhibit antibacterial activity, they were able to significantly inhibit biofilm formation in three of the four reference strains used in this study. While both **5** and **6** inhibited biofilm formation in *Escherichia coli* ATCC 25922 and *Staphylococcus aureus* ATCC 29213, only **5** inhibited biofilm formation in *E. faecalis* ATCC 29212. Interestingly, **6** exerts more extensive effect, displaying the strongest inhibitory activity in *S. aureus* ATCC 29213. In summary, secondary metabolites isolated from this marine-derived fungus are more preponderant in antibacterial and antibiofilm activities than anticholinesterase activity.

## Figures and Tables

**Figure 1 marinedrugs-19-00457-f001:**
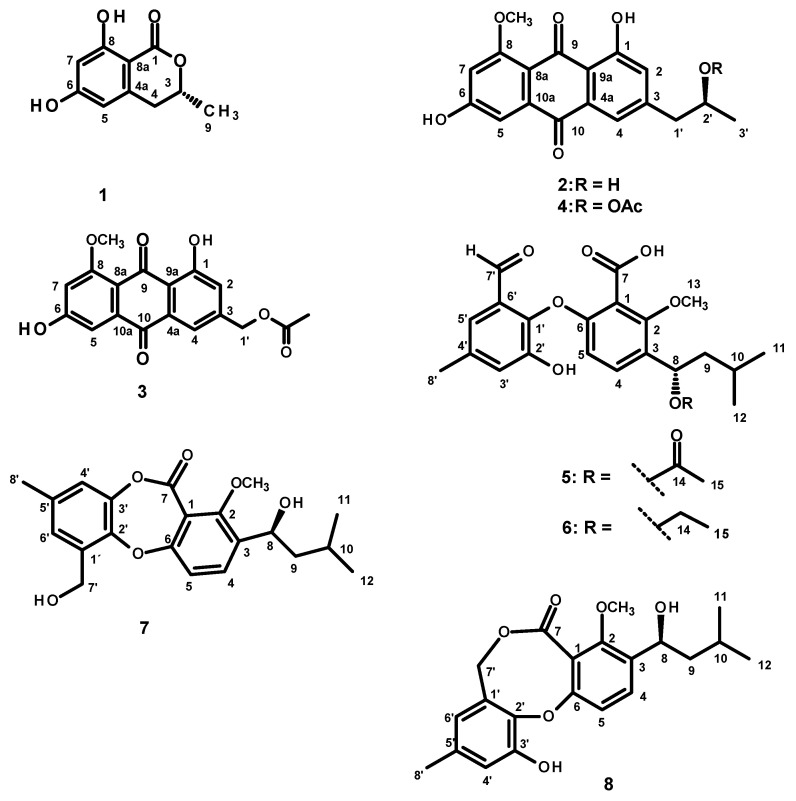
Structures of (*R*)-6-hydroxymellein (**1**), penipurdin A (**2**), acetylquestinol (**3**), acetylpenipurdin A (**4**), tenellic acid C (**5**), neospinosic acid (**6**), spinolactone (**7**), and vermixocin A (**8**).

**Figure 2 marinedrugs-19-00457-f002:**
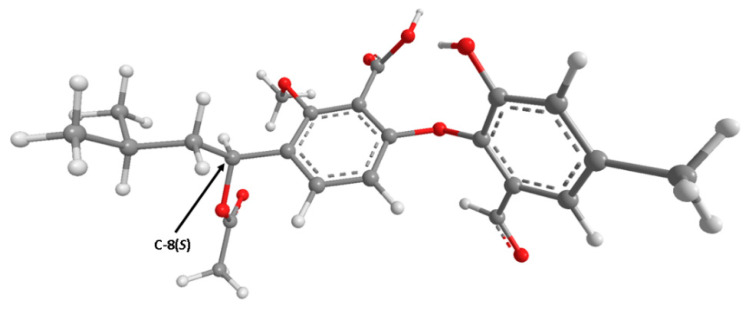
Model of one of the most abundant conformations of **5** (lowest B3LYP/6-31G/methanol energy conformer) in its ECD assigned (8*S*) configuration. Many other conformations have very similar energies to this one.

**Figure 3 marinedrugs-19-00457-f003:**
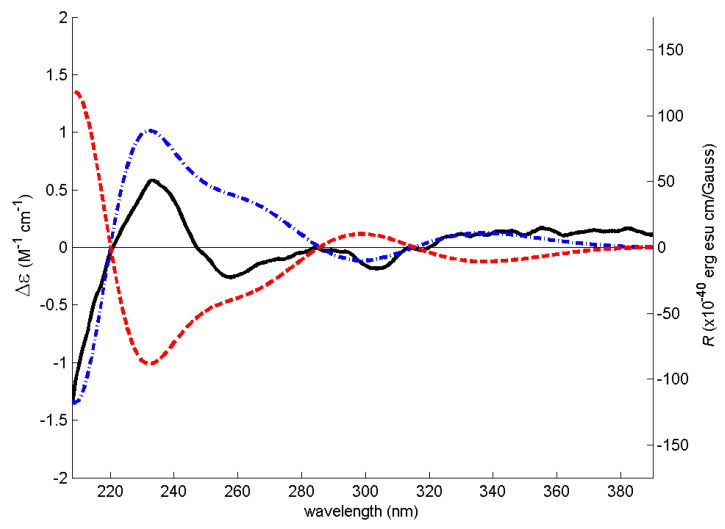
Experimental methanol ECD spectrum of **5** (solid black line) and theoretical ECD spectra of its (*S*) (dot–dashed blue line) and (*R*) (dashed red line) computational conformers.

**Figure 4 marinedrugs-19-00457-f004:**
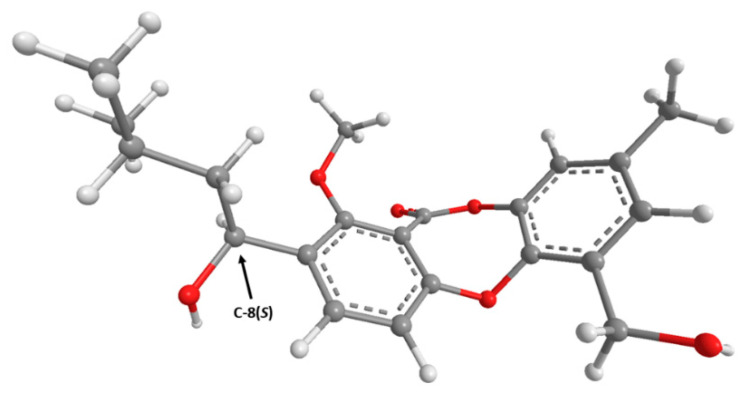
Model of the most abundant conformation of **7** (lowest B3LYP/6-31G/acetonitrile energy conformer) accounting for 48% of conformer population) in its ECD assigned (8*S*) configuration.

**Figure 5 marinedrugs-19-00457-f005:**
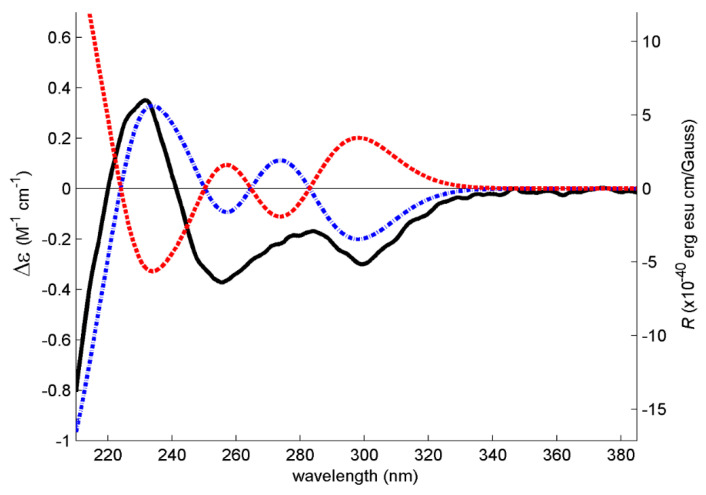
Experimental ECD spectrum of **7** in acetonitrile (solid black line) and theoretical ECD spectra of its (*S*) (dot–dashed blue line) and (*R*) (dashed red line) computational models.

**Figure 6 marinedrugs-19-00457-f006:**
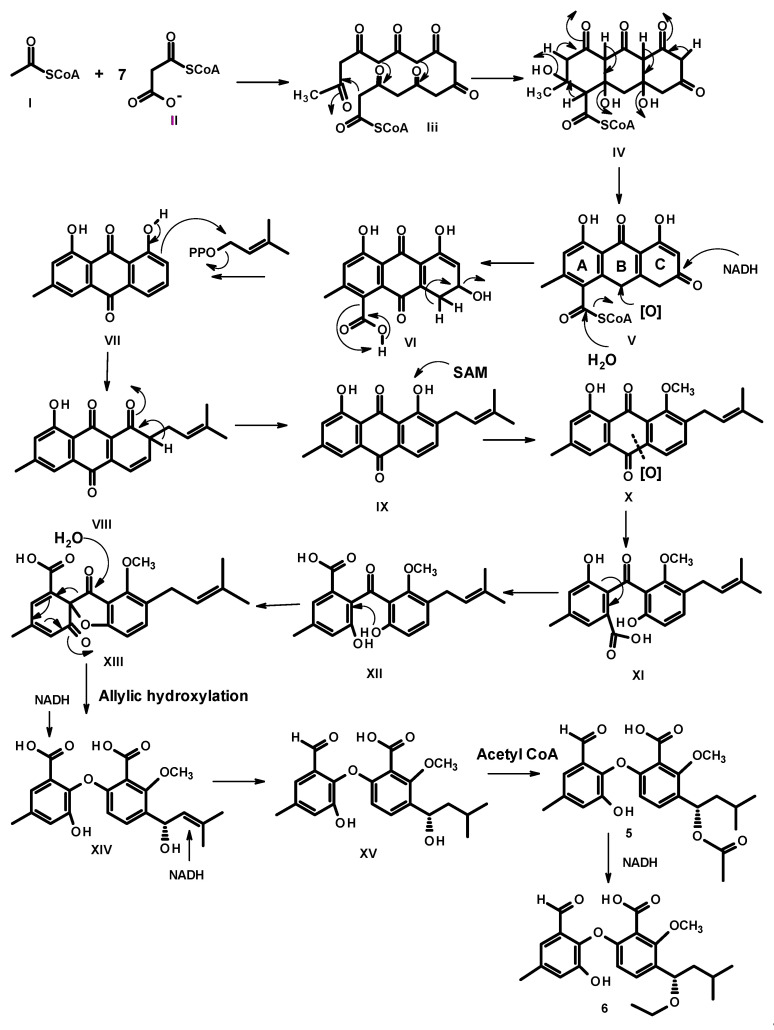
Proposed biogenesis of **5** and **6**.

**Figure 7 marinedrugs-19-00457-f007:**
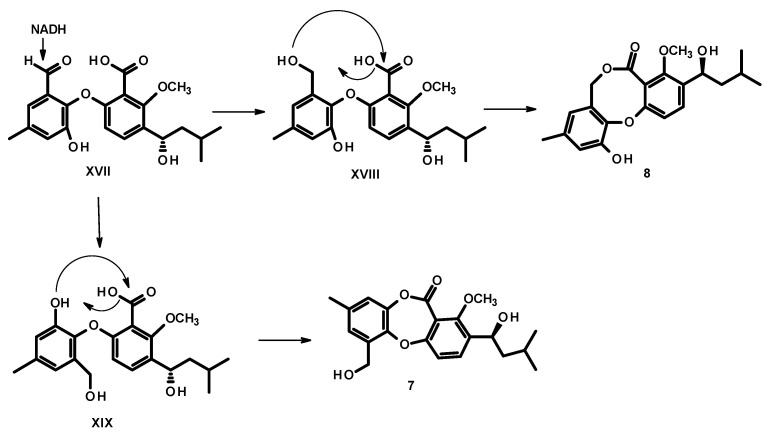
Proposed biogenesis of **7** and **8**.

**Figure 8 marinedrugs-19-00457-f008:**
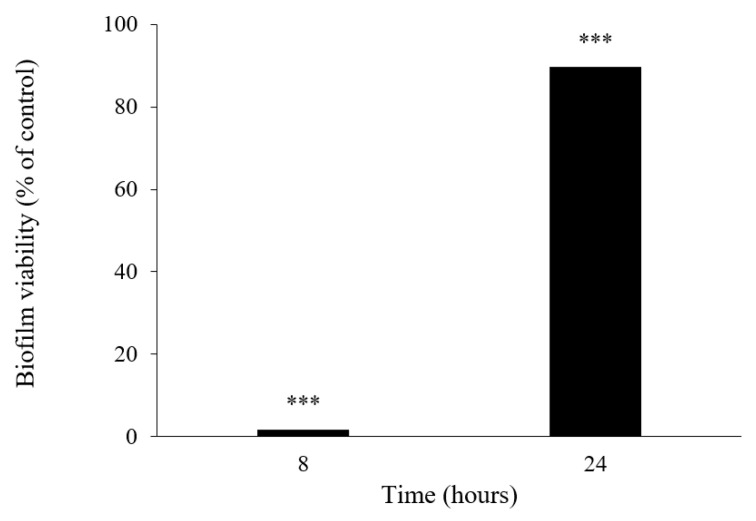
Biofilm viability effect in *S. aureus* ATCC 29213 in the presence of **6** after 8 and 24 h of incubation. Where *** represent statistical significance of data as *p* < 0.001.

**Figure 9 marinedrugs-19-00457-f009:**
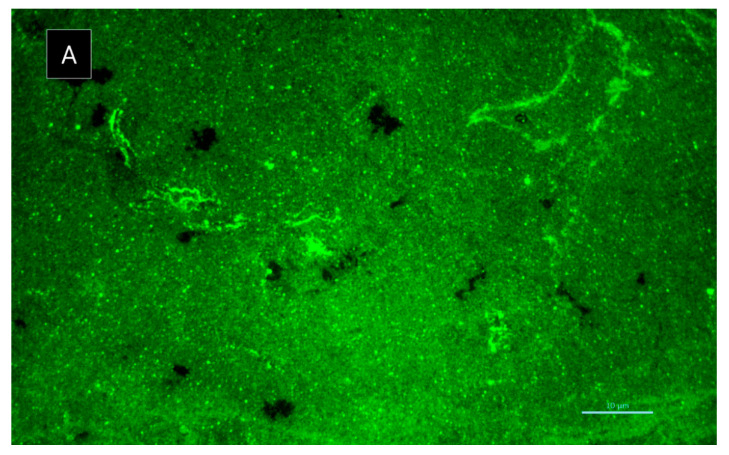
Rhodamine-conA staining of *S. aureus* ATCC 29213 biofilm: (**A**) in the absence of **6** and (**B**) in the presence of **6** after 8 h incubation. Scale bar = 10 µm. Amplification 1000×.

**Table 1 marinedrugs-19-00457-t001:** ^1^H and ^13^C NMR (DMSO-*d*_6_, 500 and 125 MHz) and HMBC assignment for **4**.

Position	δ_C_, Type	δ_H_, *J* in Hz	COSY	HMBC
1	162.1, C			
2	125.1, CH	7.15, d (1.5)	H-4	C-1, 1′, 4, 9a
3	146.6, C			
4	119.6, CH	7.47, d (1.5)	H-2	C-1′, 2, 9a, 10
4a	132.6, C			
5	108.3, CH	7.16, d (2.2)	H-7	C-7, 8a, 10
6	166.6, C			
7	105.5, CH	6.78, d (2.2)	H-5	C-5, 6, 8a
8	164.1, C			
8a	112.2, C			
9	186.4, CO			
9a	115.6, C			
10	183.0, CO			
10a	137.2, C			
OMe-8	56.7, CH_3_	3.89, s		C-8
1′	41.5, CH_2_	2.92, m	H-2′	C-2. 3, 4
2′	70.7, CH	5.06, m	H_2_-1′, H_3_-3′	C-3, CO (Ac)
3′	20.0, CH_3_	1.20, d (6.3)	H-2′	C-1′, 2′
CO (Ac)	170.2, CO			
Me (Ac)	21.4, CH_3_	1.94, s		
OH-1′	-	13.40, s		C-1, 2, 9a

**Table 2 marinedrugs-19-00457-t002:** ^1^H and ^13^C NMR (DMSO-*d*6, 300 and 75 MHz) and HMBC assignment for **6**.

Position	δ_C_, Type	δ_H_, (*J* in Hz)	COSY	HMBC
1	119.3, C	-		
2	155.1, C	-		
3	130.5, C			
4	128.5, CH	7.25, d (8.7)	H-5	C-2, 6, 8
5	110.3, CH	6.32, d (8.7)	H-4	C-1, 3, 6, 7 (w)
6	154.8, C	-		
7	167.2, CO	-		
8	73.1, CH	4.58, dd (9.2, 3.8)	H-9a, 9b	C-3, 9, 14
9a, b	47.1, CH_2_	1.57, ddd (13.8, 9.2, 5.0) 1.27, ddd (13.8, 8.9, 3.9)		C-8, 9, 11, 12
10	24.9, CH	1.71, m	H-9a, 9b, 11, 12	
11	23.7, CH_3_	0.88, d (6.6)	H-10	C-9, 10, 12
12	22.2, CH_3_	0.92, d (6.6)	H-10	C.9, 10, 11
13	62.8, OMe	3.83, s		
14	63.9, CH_2_	3.25, q (7.0)	H-15	C-8, 15
15	15.7, CH_3_	1.06, t (7.0)	H-14	C-14
1′	129.8, C	-		
2′	142.3, C	-		
3′	150.9, C	-		
4′	124.2, CH	7.14, d (2.0)	H-6′, 8′	C-2′, 3′, 6′, 8′
5′	136.7, C	-		
6′	118.4, CH	7.11, d (2.0)	H-4′, 6′	C-2′, 3′ (w), 7′, 8′
7′	189.8, CHO	10.15, s		C-1′, 5′ (w), 6′
8′	21.0, CH_3_	2.31, s	H-4′, 6′	

w = weak.

**Table 3 marinedrugs-19-00457-t003:** ^1^H and ^13^C NMR (DMSO-*d*6, 500 and 125 MHz) and HMBC assignment for **7**.

Position	δ_C_, Type	δ_H_, (*J* in Hz)	COSY	HMBC
1	114.3, C	-		
2	157.9, C	-		
3	138.3, C	-		
4	132.8, CH	7.67, d (8.6)	H-5	C-1, 3, 6
5	115.6, CH	7.21, d (8.6)	H-4	C-2, 6, 8
6	160.2, C	-		
7	161.6, CO	-		
8	64.4, CH	4.87, ddd (9.2, 4.9, 4.2)	H-9a, 9b, OH-8	
9a, b	48.3, CH_2_	1.24, ddd (13.7, 9.2, 4.2) 1.44, ddd (13.7, 9.2, 4.9)	H-8, 9b, 10 H-8, 9a, 10	
10	24.8, CH	1.72, m	H-9a, b, Me -11, 12	
11	23.9, CH_3_	0.86, d (6.7)	H-10	C-9, 12
12	22.1, CH_3_	0.90, d (6.7)	H-10	C-9, 11
13	63.1, OMe	3.76, s		C-2
1′	135.6, C	-		
2′	145.9, C	-		
3′	143.4, C	-		
4′	119.9, CH	7.11, brs		C-2′, 3′, 6′
5′	136.3, C	-		
6′	125.9, CH	7.12, d (0.5)		C-2′, 4′, 7′,8′
7′	57.9, CH_2_	4.72, d (5.8)	OH-7′	C-1′, 2′, 6′
8′	20.9, CH_3_	2.28, s		C-4′, 5′, 6′
OH-7′		5.34, t (5.8)	H-7′	C-7′
OH-8		5.13, d (4.9)	H-8	C-9

**Table 4 marinedrugs-19-00457-t004:** Antibacterial activity of **2** and **5**–**8** against Gram-positive reference and multidrug-resistant strains. MIC is expressed in µg/mL. Ceftazidime and kanamycin were used as positive controls.

Compound	*E. faecalis* ATCC 29212	*E. faecalis* B3/101 (VRE)	*S. aureus* ATCC 29213	*S. aureus* 66/1 (MRSA)
**2**	>64	>64	>64	>64
**5**	>64	>64	>64	>64
**6**	>64	>64	>64	>64
**7**	>64	64	>64	>64
**8**	>64	>64	>64	>64
**CAZ**	-	-	8	-
**KAN**	32	-	-	-

MIC, minimal inhibitory concentration. CAZ, ceftazidime. KAN, kanamycin.

**Table 5 marinedrugs-19-00457-t005:** Percentage of biofilm formation for compounds that showed antibiofilm activity after 24 h incubation.

Compound	Concentration (µg/mL)	Biofilm Biomass (% of Control)		
		*E. coli* ATCC 25922	*E. faecalis* ATCC 29212	*S. aureus* ATCC 29213
**5**	64	88.39 ± 0.09 ***	75.89 ± 0.10 ***	84.46 ± 0.10 ***
**6**	64	83.89 ± 0.19 ***	-	56.00 ± 0.06 ***

Data are shown as mean ± SD of three independent experiments. One-sample *t*-test: *** *p* < 0.001 significantly different from 100%. MIC, minimal inhibitory concentration.

**Table 6 marinedrugs-19-00457-t006:** Tyrosinase inhibitory activity of **2** and **5**–**8**.

Compounds	% Inhibition at 200 µM	IC_50_ (µM)
**2**	11.56 ± 2.05 *	n.d.
**5**	4.58 ± 0.07 ***	n.d.
**6**	n.a.	-
**7**	5.33 ± 0.18 ***	n.d.
**8**	53.1 ± 1.17 ***	177.03 ± 8.17 **
Kojic acid (positive control)	95.04 ± 0.018 ****	14.00 ± 0.12 ****

Results are given as mean ± SEM of three independent experiments performed in triplicate; n.a.: not active; n.d.: not determined; *p* < 0.05 (*); *p* < 0.01 (**); *p* < 0.001 (***); *p* < 0.0001 (****).

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
