# Peer review of "Anthraquinones, Diphenyl Ethers, and Their Derivatives from the Culture of the Marine Sponge-Associated Fungus Neosartorya spinosa KUFA 1047†"

_marinedrugs, 2021, doi:10.3390/md19080457_

Round 1

Reviewer 1 Report

De Sá et al. isolated and investigated 3 new and 5 known natural products from the marine sponge-associated fungus Neosartorya spinosa KUFA 1047. Besides elucidation of the relative and absolute configurations several biological tests were performed on the molecules.

My major concern is connected to the chiroptical studies. The authors state that they elucidated the absolute configuration of 5 and 7 unambiguously, while there were hundreds of low-energy conformers optimized first at a semiempirical level and then only c.a. 20 conformers were re-optimized at a very low DFT level of theory. Although the number of the conformers may explain a two level optimization, even the second higher level is much lower than suggested 10-20 years ago for the conformational analysis part of the ECD calculations. The basis set should be enhanced significantly. Furthermore, B3LYP was found to fail to estimate well the Boltzmann weights of the conformers in several cases. ECD reviews from the last ten years point out these issues and suggest more adequate levels for the DFT optimization. Some DFT benchmark studies would be also worth to check to find a good balance between CPU time and quality. For the TDDFT calculations, B3LYP can be a good choice but in combination with a much larger basis set. If you check the ECD literature or just a few recent reviews, you can find examples where overall good agreements led to misassignment due to the insufficient applied level of theory for the geometry optimization. Thus an “unambiguous” determination can not be done only based on an acceptable agreement between the experimental and the computed ECD spectra. If all low-energy conformers have similar ECD spectra (I found no information about the spectra of the individual conformers either in the manuscript or in the SI) than a relatively low-level might be sufficient for the calculations, but if there are conformers with nearly mirror-image spectra, which is rather common, both the DFT optimization and the ECD calculation should be performed with care at a high and well performing level or even more at least two separate combinations of levels.

To elucidate the AC of 4, the authors apply simple comparison of the specific optical rotation to relative derivatives. Although the sign of the OR is often the same in a group of related compounds, there are several exceptions described in the literature, since in OR all configurational and conformational data manifest in a positive or negative sign and thus sometimes even a slight change in the substitution pattern may result in a sign change. Consequently, the sign of a new derivative can not be applied to elucidate the absolute configuration. It can be a strong indication, but not a proof. If you want to utilize the OR for determination of the AC, the relatively large value allows safe application of OR calculations performed at similarly high level as the ECD ones should be done. On the other hand, the biosynthetic considerations (indicated by the authors) should be more solid proofs than the sign of the OR.

Minor concerns:

- For compounds containing only 1 chirality centers the numbering of the chiral carbon shall not be given together with the AC descriptor, since there are no more possibility which carbon is R or S. Consequently, (3R)-6-hydroxymellein should be just (R)-6-hydroxymellein or (8S)-5 should be (S)-5. Check for further occurrences throughout the text and the legends.

- If you say “C-2’ of 2, i.e. 2’S.” or “absolute configuration of C-8 is 8S.” it is like your name is “name X Y” and not just “X Y”. Therefore the indicated parts should be “C-2’ of 2, i.e. (S).” and “absolute configuration of C-8 is (S).” Where the AC is in parenthesis.

- What you compute during the ECD calculations are the rotatory strength (R) and the corresponding wavelength values. R values of the lowest-energy conformer or 1-2 important conformers are sometimes depicted together with the Δε as bars. However, if you apply the equation of Stephens and Harada cited in ref. 27, you get same Δε as the measured ECD. Consequently, since there are no rotatory strength values depicted on your ECD figures, you should delete the second y axis at the right.

- Concerning Fig. 9, especially in print it is hard to see anything else than black on the two pictures. I do not know how accepted it is for proof of biofilms, but changing the brightness or contrast in the same extent for both and indicating it in the legend might result in better representations of the result. By the way, you indicate A and B in the legend but the pictures are not labeled as A and B.

- On page 14 several compound numbers are not bold. Check for other occurrences.

- In the main text you write “The optical rotation of a mixture of 3 and 4 is dextrorotatory, with [α]20D + 142.8 (c 0.04, MeOH).” while in 3.3.1. you write “[α]20D = +142.8º (c 0.035, MeOH).” This is contradictory for me, since you know that “Compounds 3 and 4 were isolated as a 1:3 mixture (estimated by the integration of protons in the 1H NMR spectrum).” A specific optical rotation is a value for 1 g, thus if you have a pure compound and a mixture of the same compound and an achiral compound the specific rotation of the two can not be the same. You either have to correct with the mixture ratio in the experimental part, or give an explanation for the first sentence what that value stands for.

- A point is missing from the end of the data of 3.3.2.

- In the Acknowledgments “We Ms. Sara Cravo” should be perhaps “We thank Ms. Sara Cravo”.

Author Response

Reviewer #1

My major concern is connected to the chiroptical studies. The authors state that they elucidated the absolute configuration of 5 and 7 unambiguously, while there were hundreds of low-energy conformers optimized first at a semiempirical level and then only c.a. 20 conformers were re-optimized at a very low DFT level of theory. Although the number of the conformers may explain a two level optimization, even the second higher level is much lower than suggested 10-20 years ago for the conformational analysis part of the ECD calculations. The basis set should be enhanced significantly. Furthermore, B3LYP was found to fail to estimate well the Boltzmann weights of the conformers in several cases. ECD reviews from the last ten years point out these issues and suggest more adequate levels for the DFT optimization. Some DFT benchmark studies would be also worth to check to find a good balance between CPU time and quality. For the TDDFT calculations, B3LYP can be a good choice but in combination with a much larger basis set. If you check the ECD literature or just a few recent reviews, you can find examples where overall good agreements led to misassignment due to the insufficient applied level of theory for the geometry optimization. Thus an “unambiguous” determination can not be done only based on an acceptable agreement between the experimental and the computed ECD spectra. If all low-energy conformers have similar ECD spectra (I found no information about the spectra of the individual conformers either in the manuscript or in the SI) than a relatively low-level might be sufficient for the calculations, but if there are conformers with nearly mirror-image spectra, which is rather common, both the DFT optimization and the ECD calculation should be performed with care at a high and well performing level or even more at least two separate combinations of levels.

Reply

We thank reviewer#1 for the careful review of the ECD absolute configuration assignments. In fact, we agree with some of the reviewer#1’s concerns and that is the reason why we gave a relatively detailed account of the procedure in the Experimental Section, so that the readers are informed of the steps taken. However, due to our previous experience, we are convinced that those concerns are not strong enough to question the conclusion, and try to explain why (below), although we recognize the adverb “unambiguously” can sound too strong and can be removed from the manuscript.

In some of our previous (published) works, when we had smaller chiral molecules than compounds 5 and 7 to work with, we sometimes took the opportunity to compare the calculation results of a semi-empirical method (PM6) and two DFT methods, APFD and B3LYP, coupled with two or three different basis sets, from 6-31G to 6-311+G(2d,p). This was done as a sort of (non-published) internal calibration of the methods to have a notion of how much we can compromise calculation accuracy in favor of calculation speed and still have an unambiguous ECD configuration assignment when the choice is between enantiomers (not between diastereoisomers). We concluded that, when the conformational space is large and spread out in energy, entailing a longer calculation, it is more important to consider as much conformers as possible, using a lower level of calculation, than a higher level of calculation but cutting on the conformer population. As reviewer#1 rightfully pointed out, conformers can have wildly varying spectra. We also noticed that the energy ranking of the conformers change with the calculation’s level of accuracy but not so drastically that the Boltzmann-weighed final spectra are modified enough to preclude a correct enantiomeric assignment (taking as correct the assignment made with APDF/6-311+G(2d,p)). Of course, extrapolating from these observations to larger, albeit of the same kind, molecules is always a risk but we have some evidence that we think supports doing it.

Taking the example of compound 5. Since it is a relatively large molecule, the SCF/thermal/ECD calculation of a single conformer at the B3LYP/6-31G/IEFPCM level takes us between half-day to a day. Unfortunately, 5’s 351 MM2 different conformers had an almost linear distribution of MM2 energies with the most stable conformer representing only a bit less than 3% of the population. Therefore, we chose to calculate a PM6/IEFPCM semi-empirical energy ranking of all 351 conformers instead of drastically cut this population (risking leaving important conformers behind) to be able to use a more accurate method in reasonable time. We confronted a similar situation with compound 7.

Reviewer #1

To elucidate the AC of 4, the authors apply simple comparison of the specific optical rotation to relative derivatives. Although the sign of the OR is often the same in a group of related compounds, there are several exceptions described in the literature, since in OR all configurational and conformational data manifest in a positive or negative sign and thus sometimes even a slight change in the substitution pattern may result in a sign change. Consequently, the sign of a new derivative can not be applied to elucidate the absolute configuration. It can be a strong indication, but not a proof. If you want to utilize the OR for determination of the AC, the relatively large value allows safe application of OR calculations performed at similarly high level as the ECD ones should be done. On the other hand, the biosynthetic considerations (indicated by the authors) should be more solid proofs than the sign of the OR.

Reply:

We wish to thank reviewer#1 for raising this subject about determination of the absolute configuration of two diferent compounds by comparison of their optical rotations. In fact, we are well aware of this matter. Therefore, we modified the wording of this statement to “Based on the biogenetic consideration, the absolute configuration of C-2’ in 4 should be the same as that of C-2’ in 2, i.e. (S). This hypothesis was supported by the same sign of their optical rotations (dextrorotatory) since Singh et al. [17] have reported that 1,3,8-trihydroxy-6-(2’-acetoxyptopyl) anthracene-9,10-dione, isolated from the marine crinoid Pterometra venusta, ([α]25D + 40, c 0.05, MeOH) and its deacetylated product ([α]25D + 37, c 0.05, MeOH) were both dextrorotatory.”  As reviewer #1 can see that we did not determine the absolute configuration of C-2’ of compound 4. We only suggested that the absolute configuration of C-2’ should be (S), based on two hypotheses. The first hypothesis is biogenic consideration and the second hypothesis is the analogy with the same signs of the optical rotation of 3,8-trihydroxy-6-(2’-acetoxyptopyl) anthracene-9,10-dione and its deacetylated derivative as reported by Singh et al. [Ref 17]. As we can see that 1,3,8-trihydroxy-6-(2’-acetoxyptopyl) anthracene-9,10-dion is also a hydroxylated anthraquinone, differing from compound 4 only the substitution on C-8 (a methoxy group instead of a hydroxyl group in compound 4). The analogy of 1,3,8-trihydroxy-6-(2’-acetoxyptopyl) anthracene-9,10-dione and its deacetylated derivative is an additional support of the biogenic consideration to hypothesize that the absolute configuration of C-2’ of compound 4 should be the same as that of penidurdin A (2).

Minor concern:

Reviewer #1

 For compounds containing only 1 chirality centers the numbering of the chiral carbon shall not be given together with the AC descriptor, since there are no more possibility which carbon is R or S. Consequently, (3R)-6-hydroxymellein should be just (R)-6-hydroxymellein or (8S)-5 should be (S)-5. Check for further occurrences throughout the text and the legends.

Reply:

We wish to thank reviewer #1 for calling attention in this aspect and we do agree with reviewer #1’s suggestion. We have modified the nomenclature by removing the number of the stereogenic carbon where we find it appropriate.

Reviewer #1

If you say “C-2’ of 2, i.e. 2’S.” or “absolute configuration of C-8 is 8S.” it is like your name is “name X Y” and not just “X Y”. Therefore the indicated parts should be “C-2’ of 2, i.e. (S).” and “absolute configuration of C-8 is (S).” Where the AC is in parenthesis.

Reply:

We do agree with reviewer #1’s suggesion and we have modified all the designations of the absolute configuration accordingly.

Reviewer #1

What you compute during the ECD calculations are the rotatory strength (R) and the corresponding wavelength values. R values of the lowest-energy conformer or 1-2 important conformers are sometimes depicted together with the Δε as bars. However, if you apply the equation of Stephens and Harada cited in ref. 27, you get same Δε as the measured ECD. Consequently, since there are no rotatory strength values depicted on your ECD figures, you should delete the second y axis at the right.

Reply:

The right y axis is intended for a bar-style plot of the R values given by Gaussian (but Boltzmann-weighed). This in fact is not shown because past reviewers of Marine Drugs have requested its removal from the plot (because it clutters the image), but the axis stayed on. While we think the R axis gives a notion of the calculated transition intensities, it can be removed.

Reviewer #1

Concerning Fig. 9, especially in print it is hard to see anything else than black on the two pictures. I do not know how accepted it is for proof of biofilms, but changing the brightness or contrast in the same extent for both and indicating it in the legend might result in better representations of the result. By the way, you indicate A and B in the legend but the pictures are not labeled as A and B.

Reply:

We acknowledge reviewer#1’s concern and observation. Indeed, the quality of the file was quite poor and impaired viewing. Therefore, we have edited both pictures in the NIS software saved them in TIFF format in order to deliver the correct image quality. Both images were embedded in the body of the text in a revised manuscript.

Reviewer #1

On page 14 several compound numbers are not bold. Check for other occurrences.

Reply:

We thank reviewer #1 for detecting these typos. We have put all the number of the compounds throughout the text in “bold”.

Reviewer #1

In the main text you write “The optical rotation of a mixture of 3 and 4 is dextrorotatory, with [α]20D + 142.8 (c 0.04, MeOH).” while in 3.3.1. you write “[α]20D = +142.8º (c 0.035, MeOH).” This is contradictory for me, since you know that “Compounds 3 and 4 were isolated as a 1:3 mixture (estimated by the integration of protons in the 1H NMR spectrum).” A specific optical rotation is a value for 1 g, thus if you have a pure compound and a mixture of the same compound and an achiral compound the specific rotation of the two can not be the same. You either have to correct with the mixture ratio in the experimental part, or give an explanation for the first sentence what that value stands for.

Reply:

We agree with reviewer #1’s comment. The correct value of the mixture of 3 and 4 is, in fact, “[α]20D = +142.8º (c 0.035, MeOH)” and not “ [α]20D + 142.8 (c 0.04, MeOH).” So, we have corrected this value in the main text and deleted the value of [α]20D of compound 4 in the Experimental (3.3.1. Acetylpenipurdin A (4)). We don’t think it is correct to deduce the value of [α] D by using the ratio of the intergration of the protons in the NMR as this ratio is estimated and not of absolute precision.

Reviewer #1

A point is missing from the end of the data of 3.3.2

Reply:

Done

Reviewer #1

In the Acknowledgments “We Ms. Sara Cravo” should be perhaps “We thank Ms. Sara Cravo”.

Reply:

The correction is made.

Reviewer 2 Report

A file attached.

Author Response

Reviewer #2

We wish to thank reviewer #2 for his/her suggestions and comments:

In the introduction, the citation of the author’s previous studies is not quite necessary. Lines 50-56 may be removed.

Reply:

We think our brief mention of our previous stdies of members of the genus Neosartorya from marine sources in the introduction is fundamental to show the structural diversity of secondary (specialised) metabolites isolated from several members of this genus, which in turn, is our motivation/reason to study a marine-derived N. spinosa KUFA1047. So, we keep this paragraph in the Introduction.

Reviewer #2

In figure 1, the individual structures with the same carbon framework may be presented in a single structure. For example, compounds 2 and 4 may be presented in a single carbon framework as follows. Compounds 5 and 6 may also be presented in a single carbon framework.

Reply:

We thank reviewer #2 for his/her suggestions with which we completely agree. Therefore, we have reformulated the structures in Figure 1 as suggested by reviewer #2.

Reviewer #2

Lines 92-138: the structure elucidation of compound 4 is in too much redundant detail. Compound 4 is just an acetate of compound 2. Description of the spectral data that indicate the key difference compared to compound 2 would be sufficient. The paragraphs of lines 92-138 may be substantially trimmed.

Reply:

It is true that compound 4 is the acetate of compound 2. Since there is no discussion of the structure elucidationn of compound 2, we maintained the part of the discussion of compound 4 which we described the signals and types of the protons and carbons found in its 1H and 13C NMR spectra to conclude that these data resembled those of compound 2. We think that just saying the structure of compound 4 differs from that of 2 by the presence of the acetyl group will compromise a readability of this manuscript. However, we have removed a discussion of COSY and HMBC correlations from line 116-122.

Reviewer #2

Line 131: It would be better to cite the optical rotation value of compound 2.

Reply:

We do agree with reviewer’s 2 suggestion and we have added the value of the optical rotation of compound 2 that we measured together that from the literature.

Reviewer #2

Figure 2 may be moved to supplementary information.

Reply:

We disagree with reviewer #2’s suggestion to move Figure 2 to SI. We think the model of one of the most abundant conformations of compound  5 we used to calculate its ECD spectrum in Figure 2  is very important for readers not only to understand the context we discussed therein but also to comprehend how to obtain the theoretical ECD spectrum of compound 5 in Figure 3. Moreover, Figure 2 also helps clarify the comment raised by reviewer#1. Therefore, we maintain Figure 2 in the main text like we have done in several of our papers in Marine Drugs and other journals.

Reviewer #2

Lines 158-210: Compound 6 is an ether analogue of compound 5. Description of only the spectral data that indicate the key difference compared to compound 5 would be sufficient. The paragraphs of lines 158-210 may be substantially trimmed.

Reply:

We thank reviewer#2 for his/her suggestion. However, we do not have the same point of view. Although the structure of compound 6 is very similar to that of compound 5, we cannot compare key difference between the two compounds without previous deduction of the 1D and 2D NMR as we did not have prior discussion of the structure of compound 5. Therefore, we will maintain the discussion of compound 6 which is a new compound to facilitate its readability.

Reviewer #2

Lines 208-209: please cite the optical rotation values of compounds 5 and 6.

Reply

We partially agree with reviewer’s suggestion. Therefore, we added the value of the optical rotation of compound 5 in the text but noto f compound 6 as the optical rotations of compounds 6 and 7 are already in the experimental section-3.32. and 3.3.3.

Reviewer #2

Figure 4 may be moved to SI.

Reply

We disagree with reviewer#2 ’s suggestion for the same reason that we have explained why we don’t move Figure 2 to SI. So, we maintain Figure 4 in the main text.

Reviewer #2

Lines 284-308: Figures 6 and 7 are self-explanatory. Therefore, lines 284-308. may be replaced with a couple of sentences.

Reply

We understand reviewer#2’s concern, however, we don’t share reviewer#2’s point of view. Figures 6 and 7 are not self-explanatory. They show mechanisms of proposed biogenic pathways and that why they need an explanation. We cannot explain the biogenic pathways of four compounds with diverse routes with a few sentences as suggested by reviewer#2. Besides, we have followed the model of our previous papers we have published in several journals, including Marine Drugs.

Reviewer #2

Figure 6: A methyl group is missing in the structure of V.

Reply

We thank reviewer#2’s sharp eyes in detecting a missing methyl group of V in Figure 6. We have now corrected the structure.

Reviewer #2

In the experimental section, 3.2. fungal material, 3.3. extraction and isolation, 3.4. electronic circular dichroism part may be substantially trimmed.

Reply

These experimental sections are very important to give information that can allow other researchers to repeat the experiment. However, we have deleted line 398-403 in section- 3.2. fungal material while maintaining the integral part of sections 3.3. extraction and isolation, 3.4. electronic circular dichroism as all the data therein are real. If we trimmed it, we would have omitted and falsified our data which we considered very wrong. Moreover, the details in “Section 3.4. electronic circular dichroism” are very important to clarify the comments made by reviewer#1.

Therefore we maintain the contente of these two sections in conformity with those in the section of antibacerual, anti-cholinesterase and anti-tyrosinase activities assays.

Reviewer #2

Figure 9 was not visible in the PDF file for review

Reply

We acknowledge reviewer#2’s concern and observation. Indeed, the quality of the file was quite poor and impaired viewing. Therefore, we have edited both pictures in the NIS software saved them in TIFF format in order to deliver the correct image quality. Both images were embedded in the body of the text in a revised manuscript.

Reviewer #2

It would be helpful if positive controls are included in tables 4 and 5, like table 6.

Reply

I presume that reviewer#2 misunderstood the nature of the assays in Table 4 (Antibacterial activity of 2 and 58 against Gram-positive reference and multidrug-resistant strains. MIC is expressed in µg/mL.), Table 5 (Percentage of biofilm formation for compounds that showed antibiofilm activity after 24 h incubation) and Table 6 (Tyrosinase inhibitory activity of 2 and 58.).

Table 4 is the assay of antibacterial activity of the isolated compounds to determine their minimum inhibitory concentration (MIC). Therefore, there is no positive control like an enzyme assay in Table 6. The control of the antibacterial activity of the compound is done without the compound which is already explained in the experimental part. This also applies to Table 5 which determines the inhibition of biofilm formation of the compounds. Therefore, there is no positive control for this assay. The control is the assay without the compound. On the contrary, Table 6 and Table 7 which determine the inhibition of the enzymes acetylcholinesterase and tyrosinase of the isolated compounds, these assays need positive controls which are galantamine (for anti-AchE) and kojic acid (for anti-tyrosinase).

Reviewer #2

Figure 8: The inhibitory effect of compound 6 on viability of S. aureus was dramatically decreased at 24 hrs. A reasonable discussion on this activity profile would be helpful.

Reply

We thank reviewer #2’s pertinent and helpful suggestion. Accordingly, we have added a possible explanation for the decrease in the inhibitory effect of the compound.

Round 2

Reviewer 1 Report

The authors improved / corrected several points raised by the reviewer, except for the most time-demanding step, do a thorough conformational analysis and ECD calculation. Yes, I agree that to re-do all the necessary calculations would require a lot of time. For larger flexible molecules we often compute for months on a supercomputer but that is the price one should pay for a reliable structure elucidation with chiroptical methods. As I wrote previously, if the low-energy conformers give similar spectra, the applied level is not that important, but if their spectra are rather different, the high level is a key point even for exponentially more CPU time.

I appreciate the author’s effort and understand their need for publication, so I accept the manuscript. On the other hand, I am very sad that the level of the ECD studies decrease in the last years by becoming a standard tool in natural product chemistry and tried to perform as quick as possible by many groups new in the chiroptical calculation field. We could also assume that a kitchen scale is perfect also for chemical use since it is applied so widely in the world and really, it is cheap and good enough for cooking and for measuring hundreds of grams or a few kilograms. However, if you want to apply it for gram or milligram scale, which it is not designed for, of course, it will not perform well. Hopefully after a few years of calculation and some misassignment you will understand why it is so important to apply a high level even at a high computational cost.

Author Response

Reply

We thank reviewer #1 for understanding our explanation. We understand reviewer #1's concern about the rigour of calculation of the ECD spectra. We have aldso reviewed several manuscripts submitted to Marine Drugs using wrong or inappropriate methodology. 

As we have explained in round 1 about our methodology and we are convinced of a validity of our method as the absolute configuration of the stereogenic carbon of compounds 5 and 7 we have determined  is the same as that of the stereogenic carbon of the biphenyl ether derivatives of the same family that share the same biosynthetic origin. 

Reviewer 2 Report

a file attached.

Author Response

Please see our reply in attachment.
